# Cer-Eval: Certifiable and Cost-Efficient Evaluation Framework for LLMs

## Abstract

As foundation models continue to scale, the size of trained models grows exponentially, presenting significant challenges for their evaluation. Current evaluation practices involve curating increasingly large datasets to assess the performance of large language models (LLMs). However, there is a lack of systematic analysis and guidance on determining the sufficiency of test data or selecting informative samples for evaluation. This paper introduces a certifiable and cost-efficient evaluation framework for LLMs. Our framework adapts to different evaluation objectives and outputs confidence intervals that contain true values with high probability. We use "test sample complexity" to quantify the number of test points needed for a certifiable evaluation and derive tight bounds on test sample complexity. Based on the developed theory, we develop a partition-based algorithm, named *Cer-Eval*, that adaptively selects test points to minimize the cost of LLM evaluation. Real-world experiments demonstrate that Cer-Eval can save 20% to 40% of test points across various benchmarks, while maintaining an estimation error level comparable to the current evaluation process and providing a 95% confidence guarantee.

## 1 Introduction

In recent years, large-language-models (LLMs) have exhibited astonishing capabilities in natural language processing. Evaluating LLMs in terms of their performance and trustworthiness, therefore, is crucial for understanding their strengths and limitations, guiding their development, and ensuring responsible deployment (Chang et al., 2024). Numerous benchmark datasets have been created to assess different aspects of LLM performance. For example, the Massive Multitask Language Understanding (MMLU) dataset (Hendrycks et al., 2021a) evaluates the knowledge and problem-solving abilities of LLMs across multiple fields such as elementary math, law and history, identifying areas in which an LLM is inferior to humans; TrustGPT (Huang et al., 2023) is proposed to assess the potential of LLMs generating toxic or hateful contents, while PromptBench (Zhu et al., 2023) and MMDT (Xu et al., 2025) test the vulnerability of LLMs to adversarial prompts that could lead to misleading or unsafe responses.

Despite the increasing number of benchmark datasets, little attention has been paid to the evaluation process itself. The current practice, which we call the static evaluation process, is simply reporting the average score over the entire test dataset. This is the method used by widely adopted platforms such as Gen AI on Vertex AI platform by Google, the open LLM leaderboard hosted by Huggingface, and the Evals framework by OpenAI.

However, this static evaluation approach has two major drawbacks. First, it does not quantify or guarantee the reliability of the result. Here, reliability means how close the evaluation result is to the truth and how confident its conclusion is. In particular, there are two sources contributing to the uncertainty in the evaluation results: the randomness in the model responses, and the randomness in the dataset used for evaluation. The lack of reliability imposes difficulty in drawing a trustworthy conclusion and further tasks such as model comparison.

Second, it is not sample-efficient and does not adapt to various evaluation scenarios. The static evaluation has to evaluate all test points, making it expensive and time-consuming, as LLMs typically have a numerous number of parameters. However, in many cases, evaluating a subset of the dataset would suffice to reach a reliable conclusion. For example, if an LLM consistently performs poorly on a randomly selected subset of a question-answering (QA) benchmark, we can confidently conclude

that this model's QA capability is below random guessing. Moreover, for users who want to evaluate the model in a dynamic and evolving manner, the reliability of the static evaluation process is further compromised. For instance, when new data points are introduced over time, we will need to re-evaluate the model to accurately reflect its performance. However, for a static process, the chance of drawing at least one wrong conclusion will approach one with repeated evaluations.

To address these challenges, we focus on two fundamental but underexplored problems in LLM evaluation: for a given LLM, test dataset, and evaluation metric,

(P1) How to design an algorithm that adapts to different evaluation scenarios and goals and provides a certifiable guarantee for its result?

(P2) How to strategically select test points to minimize evaluation cost while achieving a desired conclusion, and what is the minimum number of test points needed?

To answer these questions, we propose a certifiable online evaluation process that sequentially refines evaluation results until a user-defined estimation error and confidence level is reached, e.g., the difference between the estimation and true performance is below 0.01 with 95% probability; otherwise, the algorithm will notify the user that additional data points are needed for the desired estimation error and confidence level. Beyond early stopping, our approach reduces evaluation costs by strategically selecting test points. We propose and study a concept named *test sample complexity*, which quantifies the minimal number of test points needed to achieve an accurate and confident conclusion. Inspired by the analysis of test sample complexity, we develop an online evaluation algorithm, Cer-Eval, which dynamically partitions the input space into regions of low variance and high probability mass. This allows the evaluation to focus on informative test points, significantly reducing the number of samples needed. Our contributions are summarized as follows:

**1.** We introduce an online evaluation framework for LLMs that provides statistical guarantees on evaluation results. Unlike static evaluation, our approach applies to various evaluation goals and ensures validity within a user-specified estimation error and confidence level.

**2.** We propose the concept of test sample complexity, which characterizes the number of test points required to evaluate a model to a desired level. Both upper and lower bounds are established for test sample complexity when the only assumption is a bounded loss function. We also show that test sample complexity can be greatly reduced when certain distributional assumptions hold.

**3.** Based on the developed theory, we propose Cer-Eval (outlined in Figure 1), an adaptive evaluation algorithm that minimizes the evaluation cost through early stopping and dataset partitioning. Compared to the static evaluation baseline, Cer-Eval uses only $30\% \sim 50\%$ data points and achieves a comparable evaluation accuracy in our simulation studies. When applied to real-world benchmarks (MMLU, AlpacaEval, and MATH) to evaluate GPT-4o, Cer-Eval achieves the same evaluation accuracy using only $60\% \sim 80\%$ of the test data.

The rest of paper is organized as follows. Section 2 introduces the related literature. Section 3 formulates the problem setup and defines test sample complexity. Section 4 provides test sample complexity bounds for general cases, while Section 5 presents how to improve those bounds when distributional assumptions on the model and task are met, and proposes adaptive evaluation algorithms motivated by the developed theory. Section 6 conducts extensive experiments on the proposed algorithm compared to the baseline. We conclude the paper in Section 7.

## 2 RELATED WORK

**LLM evaluation.** Existing literature of LLM evaluation primarily focuses on (1) discussing what aspects of LLM capability should be evaluated (Liu et al., 2023; Chang et al., 2024; Gao et al., 2024) and (2) proposing appropriate datasets and criterion to assess LLM performance (Hendrycks et al., 2021a; Lin et al., 2022; Chiang et al., 2024; Zhang et al., 2024b; Dubois et al., 2024; Chen et al., 2025; 2024a;b). However, relatively little attention has been given to the evaluation process itself. The standard practice for LLM evaluation remains simple: computing the average score over a test dataset based on a selected evaluation metric. Recently, (Miller, 2024) proposed to add error bars to quantify evaluation uncertainty, and (Chiang et al., 2024) constructs an approximate confidence level using bootstrapping. Nevertheless, these methods are empirical and lack valid statistical guarantees in finite-sample scenarios. As a result, the reliability of current evaluation practices is not formally ensured. Addressing this gap is one of the key contributions of our work.

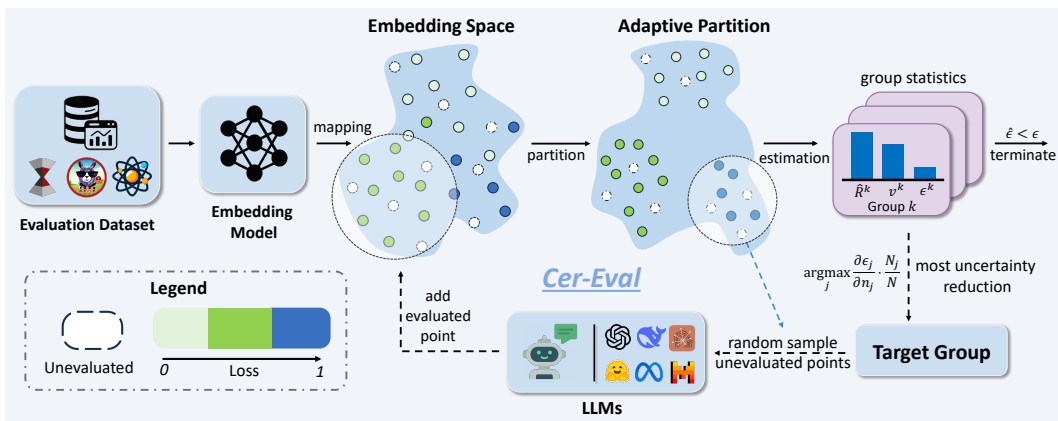

Figure 1: Overview of Cer-Eval, a partition-based adaptive evaluation algorithm. It iterates through four steps until the termination condition is met: (1) partition the dataset based on the evaluated points to minimize evaluation uncertainty, (2) compute summary statistics for each partition, (3) identify the partition that reduces uncertainty the most, and (4) sample and evaluate a new test point from the selected partition.

**Efficient evaluation.** Researchers have recognized that evaluating LLMs on the full dataset can be computationally expensive. To mitigate this, (Polo et al., 2025; Kipnis et al., 2024; Xu et al., 2024) proposed to choose a representative subset of data points to approximate the full evaluation result; (Zhang et al., 2024a) proposed to predict model performance using historical performance trends of similar models and tasks; and (Boyeau et al., 2024; Fisch et al., 2024) adopted a stratified sampling technique (Cochran, 1977) to improve evaluation accuracy, sharing a similar spirit to our partition algorithm. However, (Boyeau et al., 2024; Fisch et al., 2024) required a known partition in advance and needed to evaluate all data points. Moreover, these approaches lack formal guarantees on their evaluation results. In contrast, by leveraging the properties of the model being evaluated, Cer-Eval can adaptively find a partition and save evaluation cost by early stopping, leading to efficient evaluations with guarantees.

**Sequential hypothesis testing.** Our proposed evaluation process involves a sequential selection of test points and decision making, and we certify our evaluation results by constructing a sequence of confidence intervals (CI) covering the truth with high probability. The problem of constructing valid sequential CIs has been studied in the literature (Farrell, 1964; Karp & Kleinberg, 2007; Zhao et al., 2016; Waudby-Smith et al., 2024), with key techniques relying on Hoeffding-type concentration inequalities and a union-bound argument. However, existing methods do not incorporate model- and dataset-specific structure, which we find crucial for improving evaluation efficiency. By incorporating a Bernstein-type inequality and a partition-based approach, we prove that the needed test sample size can be greatly reduced under certain conditions.

## 3 PROBLEM FORMULATION

Given a trained LLM $f : X \in \mathcal{X} \to Y \in \mathcal{Y}$, we aim to evaluate its performance on a given task. We assume that the input space $\mathcal{X} = \mathbb{R}^{d_{\text{in}}}$ consists of the embedding vector of tokens with maximum length $d_{\text{in}}$, while the output space $\mathcal{Y}$ is general. The model $f$ can be either deterministic or non-deterministic. Throughout this paper, we consider a fixed evaluation task characterized by a joint distribution $P_{XY}$ over $(X, Y)$. The model performance in task $P_{XY}$ is quantified by its prediction error $R(f, P_{XY}) := \mathbb{E}_{(X,Y)\sim P_{XY}}\ell(f(X), Y)$ for some loss function $\ell$. Examples of $\ell$ include the zero-one loss for multiple-choice tasks, similarity-based metrics for natural language understanding tasks, or human- or LLM-based scores for reasoning tasks. For notational simplicity, we denote $R(f, P_{XY})$ as $R$ when there is no ambiguity.

In practice, model performance is often assessed on a given test dataset $D_n = \{(X_i, Y_i), i = 1, \ldots, n\}$. In this case, we assume that $D_n$ represents the underlying task distribution $P_{XY}$. That is, data points in $D_n$ are independently and identically (IID) drawn from $P_{XY}$. A concrete example of LLM evaluation is provided below.

**Example 3.1.** *Massive Multitask Language Understanding (MMLU) dataset (Hendrycks et al., 2021a) includes more than 15,000 multiple choice questions covering 57 subfields. Each question*

*has four answer choices, with only one being correct. Researchers evaluate LLMs' natural language understanding capability on this dataset by averaging the prediction accuracy across all subfields. In this case, the output space is $\{1, 2, 3, 4\}$, and a zero-one loss is used as the evaluation metric. It is found that a baseline human achieves an average accuracy of 34.5%, and many LLMs perform near-randomly.*

**Evaluation Goal and Process.** The goal of model evaluation is to obtain an **accurate** and **confident** estimation of $R$. Here, accuracy means how close the estimation is to the truth, and confidence means the probability that our claim is correct. The user can specify a desired confidence level $1 - \delta$ ($\delta$ is also known as the failure probability) and estimation error level $\epsilon$. Two common evaluation goals are:

(1) Estimate $R$ within an error level of $\epsilon = 0.01$ with $95\%$ confidence.

(2) Determine whether $R$ exceeds a threshold (e.g., 0.5) with $95\%$ confidence.

Therefore, a certifiable evaluation algorithm should provide an estimate of $R$ along with a confidence interval (CI) of radius $\epsilon$, which is guaranteed to contain the true error $R$ with probability at least $1 - \delta$. Notably, the second goal above is equivalent to obtaining an estimate with $\epsilon$ implicitly determined by $R$ and the threshold. Moreover, users may wish to dynamically adjust $\epsilon$ during the evaluation process. The current practice, which we call a static evaluation process, requires evaluating all test data at once, described below.

**Definition 3.2** (Static evaluation process). A static evaluation process tests all data points in a test dataset $D_n$, and output a single estimated prediction error and CI.

Static evaluation is unable to handle an implicitly defined or flexible $\epsilon$. To address this issue, we propose an online evaluation framework, as defined below.

**Definition 3.3** (Online evaluation process). An online evaluation algorithm $\mathcal{A}$ sequentially selects test points until the desired evaluation error level $\epsilon$ is achieved, or all available test data points are used. The number of *evaluated* test points at termination is denoted as $N$.

**Definition 3.4** (($s, \epsilon, \delta$)-certified evaluation algorithm). An algorithm $\mathcal{A}$ is called $(s, \epsilon, \delta)$-certified if

$$\mathbb{P}(N \geq s, \text{ or } \mathcal{A} \text{ produce at least one CI that does not contain truth}) \leq \delta.$$

*Remark* 3.5 (Practical meaning of certified algorithms). For example, when the user's goal is estimating the model performance $R$, an estimation obtained from an $(s, \epsilon, \delta)$-certified algorithm is guaranteed to be within $\epsilon$ of the true value with probability at least $1 - \delta$.

**Definition 3.6** (Test sample complexity). Consider any algorithm $\mathcal{A}$ that is $(s, \epsilon, \delta)$-certified for evaluating a model $f$ on task $P_{XY}$. Test sample complexity $n^*$ is the smallest $s$ over all possible choices of algorithms $\mathcal{A}$.

A $(s, \epsilon, \delta)$-certified test algorithm provides guarantees on how confident the evaluation result is and how many test points are needed for this algorithm. Test sample complexity, which is the minimal required number of test points to draw a confident and accurate conclusion, further characterizes the fundamental difficulty of evaluating a model on a given task. The subsequent sections are dedicated to obtain bounds on the test sample complexity, and propose efficient evaluation algorithms, thereby answering the core research problems introduced in Section 1.

*Remark* 3.7. The static evaluation process can be regarded as a special online algorithm that only yields a result after evaluating all $n$ points.

## 4 INTRINSIC LIMITS OF TEST SAMPLE COMPLEXITY

In this section, we establish matching upper and lower bounds on test sample complexity, assuming only that the loss function is bounded. Theorem 4.2 indicate the fundamental limit on the number of test points needed for a certifiable evaluation across general models and tasks.

**Assumption 4.1** (Bounded loss). We assume that the loss function $\ell$ is bounded. Without loss of generality, let $0 \leq \ell(f(X), Y) \leq 1$.

**Theorem 4.2.** *Let $\epsilon > 0$ be the desired estimation error level and $0 < \delta < 1$ be the failure probability. Under Assumption 4.1, we have the following results:*

- *(Upper Bound) There exists an $(s, \epsilon, \delta)$-certified online evaluation process (Algorithm 3 in Appendix C) with $s \leq O(\epsilon^{-2}\{\ln(1/\delta) + \ln\ln(1/\epsilon)\})$*

- *(Matching Lower Bound) For any $s$ such that*

$$\lim_{\epsilon, \delta \to 0} \frac{s\epsilon^2}{\ln(1/\delta) + \ln\ln(1/\epsilon)} > 0,$$

*no algorithm can be $(s, \epsilon, \delta)$-certified for all sufficiently small $\epsilon$ and $\delta$.*

The bounds in Theorem 4.2 depend on two key parameters, the estimation error level $\epsilon$ and failure probability $\delta$. Clearly, a smaller $\delta$ requires a higher level of confidence of the evaluation, thus a larger test sample complexity is needed. A smaller $\epsilon$ demands a greater evaluation accuracy, leading to a larger test sample complexity. Compared to training sample complexity, test sample complexity focuses on a specific model and task instead of learning from a function class. Furthermore, in real-world evaluations, $\epsilon$ may be implicitly determined or dynamically adjusted. This necessitates a sequential evaluation and therefore introduces a sequential decision-making challenge, requiring the additional iterated logarithm term $\ln\ln(1/\epsilon)$ to control the overall failure probability – an effect absent in classical training sample complexity bounds.

Notably, even the vanilla online evaluation process (Algorithm 3) can significantly reduce the amount of test points compared to the static evaluation process, particularly when the desired estimation error level $\epsilon$ is not too small.

## 5 SAMPLE-EFFICIENT EVALUATION VIA PARTITION

In this section, we go beyond the intrinsic statistical bounds by incorporating additional knowledge about the model and task. To further save test points, the key idea is to pay more attention to areas with higher uncertainty, instead of drawing test points IID from the entire space. Two critical observations drive this approach: (1) An area with smaller loss variance is less uncertain and requires fewer test points for confident evaluation; and (2) Properly dividing the input space may lower the variance within each partition, reducing the number of test points needed for evaluation. Thus, if we can divide the task distribution $P_{XY}$ into $K$ disjoint areas and reduce variance within each, we can achieve a more sample-efficient evaluation than the general approach in Section 4. We formulate this idea in Theorem 5.2.

**Definition 5.1** (Benign Partition). Consider any partition $\{A_k\}_{k=1,\ldots,K}$ on the support of $P_{XY}$, and $v_k = \text{var}\{\ell(f(X), Y) \mid A_k\}$ be the variance of the loss conditioned on $A_k$. Given a test dataset $D_n = \{(X_i, Y_i), i = 1, \ldots, n\}$, let $\widetilde{D}_k = A_k \cap D_n$ and $n_k := |\widetilde{D}_k|$. We say $\{\widetilde{D}_k\}_{k=1,\ldots,K}$ is a benign partition of $D_n$ if the following holds:

$$n_k/n \geq \ln(K+1)\max\{v_k, \epsilon^{2/3}\}, k = 1, \ldots, K.$$

**Theorem 5.2.** *Suppose Assumptions 4.1 holds, and let $n = \Theta(\epsilon^{-2}\{\ln(1/\delta) + \ln\ln(1/\epsilon)\})$ denote the tight bound of Theorem 4.2 for some $\epsilon, \delta$. Then Algorithm 1 operating with a benign partition of $D_n$ (Definition 5.1) is $(n', \epsilon, \delta)$-certified, such that*

$$\rho := \frac{n'}{n} = O\left(\ln(K+1)\sum_{k=1}^{K}\max\{v_k, \epsilon^{2/3}\}\right).$$

Theorem 5.2 suggests that, with a benign partition, we can use only $\rho$ percent of test points for evaluation, compared to the vanilla evaluation algorithm in Section 4 where no additional knowledge is available. Furthermore, as $\epsilon \to 0$, allowing $K$ to grow can lead to $\rho \to 0$, significantly reducing the number of test points needed.

**Illustration via examples.** Benign partitions exist for a wide range of models and tasks, and we can have a good estimate of the saving ratio $\rho$. We illustrate it by the following corollary and example.

**Corollary 5.3** (Super-Gaussian loss distribution). *Suppose the loss distribution satisfies $h(z) := \mathbb{P}(\ell(f(X), Y) = z) \geq A\exp\{-z^2/\sigma^2\}$ for $z \in [0, 1]$ with constants $A, \sigma^2 > 0$, then, we have $\rho = O(\ln(K+1)/K)$ for any $K$ such that $K/\ln(K+1) \geq \exp(1/\sigma^2)/A$ and $K \leq \epsilon^{-1/3}$. As a result, $\rho = O(\epsilon^{1/3}\ln(1/\epsilon))$ when $K = O(\epsilon^{-1/3})$.*

**Example 5.4** (Equally distributed problem difficulty). *Consider a dataset split into $K$ difficulty levels, each having an equal number of test points. Also, the prediction loss of $f$ in $A_k$ lies in $[(k-1)/K, k/K]$ uniformly. For example, a dataset assessing LLMs on math problems may contain questions collected from primary school, high school, undergraduate, and graduate levels. Since the loss distribution is uniform on $[0,1]$, Corollary 5.3 applies, yielding $\rho = O(\ln(K+1)/K)$.*

*Remark* 5.5. The term $\max\{v_k, \epsilon^{2/3}\}$ in Theorem 5.2 arises from estimating the unknown loss variance in each partition. We can improve this term to $\max\{v_k, \epsilon\}$ if an upper bound asymptotically equivalent to $v_k$ is known. Moreover, equally-space partition is a special partition that improves this term to $1/K^2$, leading to $\rho = O(\epsilon \ln(1/\epsilon))$ in the super-Gaussian example above.

*Remark* 5.6. Our developed theory provides support for similarity-based dataset pruning methods, such as clustering. Those methods assume that the model performance is similar in a small neighborhood of any point $(X, Y)$. If the loss function is continuous, then the performance at $(X, Y)$ suffices to approximate local performance on an $\epsilon$ ball $B_\epsilon(X, Y) := \{(X', Y') : \|(X, Y), (X', Y')\| \leq \epsilon\}$ around it, reducing the number of required test points.

*Remark* 5.7. In this work, we assume test samples are IID drawn from the target distribution. When this assumption is violated, evaluation results of Cer-Eval may be biased. A possible extension is to model the distribution shift between the target $P$ and the actual test distribution $Q$. In this case, the $(\epsilon, \delta)$-evaluation-guarantee of Cer-Eval may be adjusted to $(\epsilon + dist(P, Q), \delta)$-guarantee, where $dist(P, Q)$ measures the divergence between the two distributions. Formalizing this extension is an exciting direction for future work.

**Finding effective benign partitions.** As shown in Theorem 5.2, finding a benign partition that simultaneously minimizes in-group variance and maximizes the probability mass of each group is critical in enhancing test efficiency. However, such partition information may be unavailable in practice. To address this concern, we design Cer-Eval (Figure 1), which dynamically partitions the input space in a model- and data-driven manner to maximize the benefit brought by the benign partition. Cer-Eval repeats the following steps until the desired estimation error level is achieved or all points in $D_n$ have been evaluated, with full details in Algorithm 1: (1) Adaptive partition. Partition the input space based on the evaluated points by minimizing the uncertainty level. We propose to adopt 1-nearest neighbor algorithm in the partition subroutine, as detailed in Algorithm 2. (2) Estimation. Compute the sample mean and associated CI radius for each group. (3) Target group selection. Identify the group that contributes most to uncertainty. (4) New sample evaluation. Sample and evaluate a new test point from the target group.

# 6 EXPERIMENT

## 6.1 SIMULATION

**Synthetic Data.** We generate simulated datasets as follows. First, we choose $K$ as the true number of partitions. For each partition $A_k, k = 1, \ldots, K$, the query and response pairs follow a Gaussian distribution $(X, Y) \sim N(c_k, \sigma^2 I)$, where $I$ is the identity matrix and $c_k = (\lambda k, 0, 0, \ldots) \in \mathbb{R}^d$ with some constants $d$, $\lambda$ and $\sigma^2$. The loss $\ell(f(X), Y) \mid A_k$ follows a truncated Gaussian distribution with mean $(k - 1/2)/K$ and variance $1/K^2$. In particular, we consider the following three scenarios: (S1) Single partition: $K = 1$. (S2) Multiple easy-to-distinguish groups: $K = 3$, $\lambda = 5$. (S3) Multiple hard-to-distinguish groups: $K = 3$, $\lambda = 1$. For all scenarios, we set the failure probability $\delta = 0.05$, input dimension $d = 10$, variance $\sigma^2 = 1$, and generate a test dataset of size $n = 5000$.

**Evaluation algorithms.** We compare three proposed online evaluation algorithms against a static baseline: (1) **Base**: The static evaluation process that evaluates all data points and provide a confidence interval. (2) **Seq**: The vanilla online evaluation process, detailed in Algorithm 3. (3) **Cer-Eval**: Our proposed adaptive evaluation algorithm, as described in Algorithm 1. (4) **Oracle**: A special case of Cer-Eval that uses the true partition as the partition subroutine, which is theoretically optimal.

**Evaluation metric.** We evaluate the average loss for various values of estimation error level $\epsilon \in [\epsilon^*, 0.1]$, where $\epsilon^* = \sqrt{\log(1/\delta)/(2n)}$ is the estimation error level achieved by Base. For each test algorithm and estimation error level, we report (1) the average saving ratio $\rho$, indicating the proportion of test points saved compared to 'Base', and (2) the empirical failure probability, measuring the frequency at which $R$ falls outside the computed confidence interval. The experiment is replicated 20 times in each scenario.

**Findings.** The saving ratios for all scenarios are reported in Figure 2, while the empirical failure probability is zero across all methods. Each $(\rho, \epsilon)$ pair can be regarded as a feasible solution, and a

---

**Algorithm 1** Certified Evaluation with Adaptive Partition (Cer-Eval)

---

**Require:** The estimation error level $\epsilon$, the failure probability $\delta$, test dataset $D$, warm start steps $m$, and a partition subroutine.

1: Select the first $m$ points $S = \{(X_i, Y_i), i = 1, \ldots, m\}$ from $D$    $\triangleright$ **Step 0**: Warm-up sampling
2: Evaluate $Z_i = \ell(f(X_i), Y_i), i = 1, \ldots, m$
3: **while** True **do**
4:    Partition $D$ to $K$ areas $\widetilde{D}_1, \ldots, \widetilde{D}_K$ by the partition subroutine. $\triangleright$ **Step 1**: Partition dataset
5:    **for** $k = 1, \ldots, K$ **do**        $\triangleright$ **Step 2**: Calculate summary statistics per group
6:      Let $S_k \leftarrow \{(X, Y) \in S : (X, Y) \in \widetilde{D}_k\}$
7:      Let $n_k \leftarrow |S_k|, N_k \leftarrow |\widetilde{D}_k|$        $\triangleright$ Sample size of group $k$
8:      Let $\widehat{R}^k \leftarrow n_k^{-1} \sum_{i=1}^{n_k} Z_i^k$        $\triangleright$ Empirical mean in $S_k$
9:      Let $v_k \leftarrow n_k^{-1} \sum_{i=1}^{n_k} (Z_i^k - \widehat{R}_k)^2$        $\triangleright$ Empirical variance
10:     Let $\eta_k \leftarrow \sqrt{\{2 \ln(\log(n_k) + 1) + \ln(16K/\delta)\}/n_k}$
11:     Let $\epsilon_k \leftarrow 2\eta_k^2/3 + 2\sqrt{(v_k + \eta_k + \eta_k^2)\eta_k^2}$        $\triangleright$ Confidence interval radius
12:    **end for**
13:    Let $\widehat{R} \leftarrow \sum_{k=1}^{K} N_k \widehat{R}^k / N, \widehat{\epsilon} = \sum_j N_j \epsilon_j / N$    $\triangleright$ Performance estimate and CI raduis
14:    **if** $\widehat{\epsilon} \leq \epsilon$ **then**        $\triangleright$ Termination condition
15:      Terminate and return $\widehat{R}, \widehat{\epsilon}$
16:    **end if**
17:    Terminate if all points in $D$ are evaluated
18:    $k \leftarrow \arg\max_{1 \leq j \leq K, n_j \leq |D_j|} |\frac{\partial \epsilon_j}{\partial n_j}| \cdot \frac{N_j}{N}$    $\triangleright$ **Step 3**: Target sampling, identify group that
contributes most uncertainty
19:    Select a data point $(X_j, Y_j)$ from $\widetilde{D}_k \backslash S_k$        $\triangleright$ **Step 4**: New sample evaluation
20:    Let $n_k \leftarrow n_k + 1$, add $(X_j, Y_j)$ to $S$ and evaluate $Z_{n_k}^k \leftarrow \ell(f(X_j), Y_j)$
21: **end while**
**Output:** The estimated loss $\widehat{R}$, confidence interval radius $\widehat{\epsilon}$, and number of evaluated points $\sum_k n_k$

---

**Algorithm 2** Subroutine: Partition by 1-nearest neighbor

---

**Require:** The test dataset $D$, evaluated points $S = \{(X_i, Y_i), i = 1, \ldots\}$ and the corresponding loss values $\{Z_i, i = 1, \ldots\}$.

1: **for** $k = 1, \ldots, \lceil \ln(|S|) \rceil + 1$ **do**
2:    Assign $i$-th data point $(X_i, Y_i)$ in $S$ a label $\lfloor kZ_i \rfloor, i = 1, \ldots, n$
3:    Train a 1-nearest neighbor classifier $\mathcal{C}_k$ on a random subset of S.
4:    Partition $D$ by the labels predicted using $\mathcal{C}_k$.
5:    **for** $j = 1, \ldots, k$ **do**
6:      Calculate $\epsilon_j$ following lines 6-11 in Algorithm 1
7:    **end for**
8:    $\widetilde{\epsilon}_k \leftarrow \sum_j N_j \epsilon_j / N$
9: **end for**
**Output:** $K \leftarrow \arg\min_k \widetilde{\epsilon}_k, \widetilde{D}_k$'s partitioned by $\mathcal{C}_K$

---

pair closer to the upper-left corner is preferred. The area under the $\rho$-$\epsilon$ curve therefore reflects the evaluation efficiency of an algorithm. Specifically, we have the following key observations:

- Algorithms performing variance reduction partitions significantly improve test efficiency. In the easy-to-distinguish scenario (Figure 2, middle panel), Cer-Eval and Oracle save nearly 60% of test points compared to Base for $\epsilon = 0.02$.
- Partition-based algorithms improve efficiency even for a single group. When $K = 1$, both Oracle and Cer-Eval save about 20% of test points for $\epsilon = 0.02$ and 70% for $\epsilon = 0.03$, compared to Base.
- Partition quality is crucial. Cer-Eval evaluates more efficiently in the easy-to-distinguish scenario than in the hard one, highlighting the importance of effective partitioning. Oracle is Cer-Eval with the knowledge of perfect partitions, achieving even better performance.
- Partition-based algorithms outperforms Seq in all scenarios. This is because they effectively utilize model- and dataset-specific information, while Seq does not take those information into account.

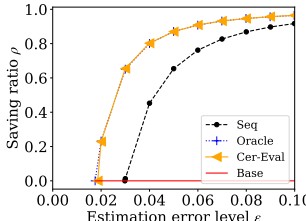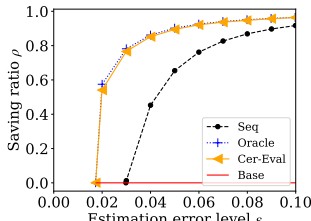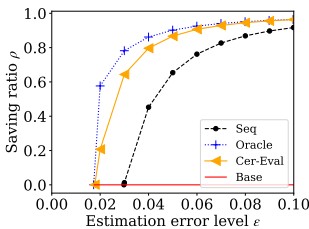

Figure 2: Percentage of test points saved by Cer-Eval compared to the baselines on **Synthetic Data** under (**Left**) the single partition scenario, (**Middle**) the easy-to-distinguish scenario, and (**Right**) the hard-to-distinguish scenario.

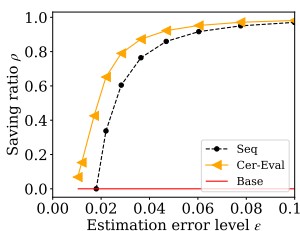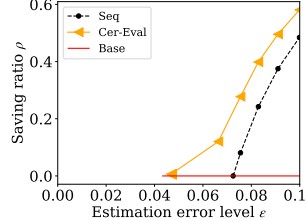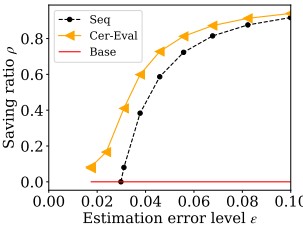

Figure 3: Percentage of test points saved by Cer-Eval compared to baselines in **Real-World Benchmarks under GPT-4o**: (**Left**) the MMLU dataset, (**Middle**) the AlpacaEval dataset, and (**Right**) the MATH dataset.

- All online evaluating algorithms guarantee the desired confidence level, successfully including the truth in the reported CI with high probability.

In short, simulation results confirm that Cer-Eval can greatly save the needed test points by adapting to the model and dataset of interest, with a controlled failure probability on the evaluation result.

## 6.2    REAL-WORLD BENCHMARKS

**Datasets.** We conduct experiments on the following three real-world datasets: (1) MMLU (Hendrycks et al., 2021a): This dataset assesses an LLM's knowledge on 14,042 multiple choice questions across 57 subjects, such as history and math. Zero-one loss is used as the evaluation metric and we are interested in the model accuracy. (2) AlpacaEval (Dubois et al., 2024): An automated evaluation benchmark that evaluates the LLM's natural language generating capability. We focus on evaluating the win rate of a target model's generated text compared to a reference model. A voting probability (or win rate score) is used as the evaluation metric. (3) MATH (Hendrycks et al., 2021b): A dataset used to measure LLMs' math problem solving abilities. We use the zero-one loss to evaluate the model accuracy.

**Algorithms, models, and results.** Since real-world datasets lack a true partition, we only compare three methods: Base, Seq, and Cer-Eval. The embedding vectors for Cer-Eval are obtained using a pre-trained BERT model (Devlin et al., 2019), with an ablation study on embedding models provided in Appendix D. The empirical failure probability is calculated as the proportion of trials where the CI does not contain the model's average performance across the entire dataset. Other experimental settings follow those of the simulation study.

We assess four models across all datasets: GPT-4o, Llama3 8B, Mistral 7B, and Qwen2 7B. Figure 3 shows the experimental result for GPT-4o. Curves for other models are similar, hence are deferred to Appendix D together with full experiment details. The empirical failure probability remains zero for all methods. We find that:

- Adaptive partition algorithm improves evaluation efficiency but varies by datasets and models. Aligning with the simulation study, the partition quality is crucial for Cer-Eval. As the partition is found adaptive to each model and task, the saving ratio of Cer-Eval thus varies. On the MATH and MMLU datasets, Cer-Eval reduces the required test samples by $30\% \sim 40\%$ for $\epsilon = 1.5\epsilon^*$ and even $5\% \sim 10\%$ for $\epsilon = \epsilon^*$. However, it achieves lower savings on AlpacaEval, reducing test points by only $10\%$ at $\epsilon = 1.5\epsilon^*$.

- Cer-Eval consistently outperforms 'Seq'. By leveraging variance information, Cer-Eval uses fewer test points than Seq and successfully identifies meaningful partitions. Notably, on both AlpacaEval

Table 1: Evaluated model accuracy on the MMLU dataset using 100 test points.

| Model | True Accuracy | p-IRT | gp-IRT | Cer-Eval |
|---|---|---|---|---|
| llama3_8b | 0.658 | 0.596 | 0.603 | 0.640 |
| mistral_7b | 0.584 | 0.518 | 0.523 | 0.620 |
| gpt_4o | 0.853 | 0.743 | 0.757 | 0.910 |
| qwen2_7b | 0.697 | 0.597 | 0.606 | 0.720 |

Table 2: Estimation error levels ($\epsilon$) achieved by StratPPI and Cer-Eval on three datasets.

| Method \ Dataset | MMLU | MATH | S3 |
|---|---|---|---|
| StratPPI | 0.013 | 0.016 | 0.010 |
| Cer-Eval | 0.010 | 0.019 | 0.018 |

and MATH datasets, there is a change point of the $\rho$-$\epsilon$ curve for Cer-Eval. For example, on AlpacaEval, the curve flattens when $\epsilon < 0.067$. A closer investigation reveals that Cer-Eval has detected two distinct data groups after this point, where the model performs well on one group and performs poorly on the other. It leads to significantly reduced within-group variance, therefore Cer-Eval obtains a confident evaluation result with fewer points. This observation aligns with our theoretical findings.

- Cer-Eval helps determine the sufficiency of test data. Note that for MMLU and MATH dataset, Cer-Eval do not evaluate all data points to achieve an estimation error level of $\epsilon^*$. It indicates that these two datasets already have sufficient data for even a smaller error level or higher confidence level. In contrast, AlpacaEval dataset has to collect more test points for a more accurate evaluation.
- All methods maintain the desired confidence level. Across all model-dataset combinations, the empirical failure probabilities remains below $0.05$, corroborating the reliability of the proposed algorithms.

**Computational overhead.** The computational overhead of Cer-Eval primarily stems from the adaptive partitioning step. However, in practice, this step is lightweight. For example, on the MMLU dataset evaluated with an 8B parameter model, the adaptive partitioning takes only a few seconds on a single H100 GPU, while full model inference takes 20 minutes. For large-scale datasets, Cer-Eval can further reduce this cost through scheduled partition updates, instead of re-partitioning after each test point. For example, we can perform partitioning after evaluating $\alpha\beta^t$ points ($\alpha > 0, \beta > 1$, $t = 1, 2, \dots$) or after every fixed percentage of the dataset. Under such schedules, the partition step is executed at most a fixed number of times $T$. As each partitioning step using the 1-NN algorithm has time complexity $O(n \ln(n))$, the overall time complexity of partitioning is $O(Tn \ln(n))$, which is almost linear in the dataset size. Thus, Cer-Eval scales efficiently to large test sets.

### 6.3 COMPARISON WITH OTHER BASELINES

**IRT-based methods.** We first compare Cer-Eval to p-IRT and gp-IRT, two item-response-theory–based methods proposed by Polo et al. (2025), which evaluate models using a small, representative subset of test points. Following their experimental setup, we ran experiments on the MMLU and AlpacaEval benchmarks using the authors' released code. We evaluate four models on MMLU, and over 200 models on AlpacaEval. In both cases, p-IRT and gp-IRT evaluate models using 100 preselected test points from Polo et al. (2025), and Cer-Eval terminates after querying 100 points.

The results, summarized in Table 1 and Figure 4, show that p-IRT and gp-IRT do not significantly outperform Cer-Eval in terms of estimation error. This is likely because the evaluated models are relatively new: p-IRT and gp-IRT require fitting an IRT model based on prior evaluation results from existing models, which may not generalize well to unseen ones. Consequently, their selected "core" test points may fail to remain representative for new models. In contrast, Cer-Eval requires no prior knowledge or pretraining on other models, making it more flexible and broadly applicable.

More importantly, Cer-Eval provides formal statistical guarantees on its evaluation accuracy, which p-IRT and gp-IRT lack. With 100 evaluated points, Cer-Eval ensures that the estimated score deviates by at most 0.12 with 95% confidence and 0.08 with 75% confidence. By contrast, IRT-based methods offer no certified error bounds, leaving practitioners uncertain about how accurate the evaluation is or whether 100 points suffice.

**Stratification-based methods.** We next compare Cer-Eval with StratPPI, proposed by Fisch et al. (2024). StratPPI assumes a known partition of the dataset and evaluates *all* test points. We thus compare them under three settings: (1) evaluate on MMLU, using "subject" as the predefined partition for StratPPI; (2) evaluate on MATH, using "difficulty level" as the predefined partition for StratPPI; and (3) evaluate on simulated dataset (S3) from Section 6.1, using the oracle partition for StratPPI.

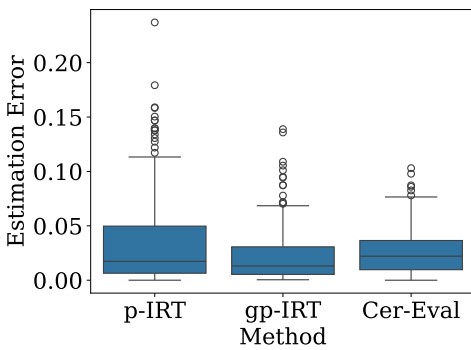

Figure 4: Estimation error on model evaluations based on 100 test points from the AlpacaEval dataset.

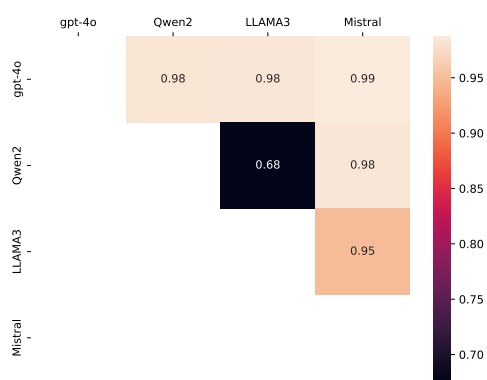

Figure 5: Saving ratios when performing pairwise comparison of model performance.

We evaluate four representative models, LLAMA3 8B, Mistral 7B, GPT-4o, and Qwen2 7B, and report the resulting estimation errors at a 95% confidence level ($\delta = 0.05$) in Table 2..

As expected, when a high-quality partition is available, StratPPI attains a slightly smaller estimation error $\epsilon$. However, when such partitions are unavailable, as is often the case in practice, Cer-Eval can adaptively construct effective partitions and sometimes outperform StratPPI, as observed on the MMLU benchmark. On MATH, Cer-Eval yields a marginally higher $\epsilon$ because it performs sequential testing, which incurs a minor $\ln! \ln(n)$ penalty to allow for early stopping. In contrast, StratPPI must evaluate every test point, even when the desired error level (e.g., $\epsilon > 0.02$, $\delta = 0.05$) could be achieved with far fewer queries.

### 6.4 EXTENSION TO COMPARE MULTIPLE MODELS

While Cer-Eval is proposed to evaluate a single model, this framework naturally extends to broader applications such as pairwise model comparison and ranking multiple models.

To illustrate this potential, we conduct an empirical demonstration of pairwise model comparison on the MMLU benchmark. We consider four models: GPT-4o, Qwen2-7B, LLaMA-3 8B, and Mistral-7B. Their oracle accuracies are 0.85, 0.70, 0.66, and 0.58, respectively, giving the ground-truth ranking: GPT-4o > Qwen2 7B > LLaMA3 8B > Mistral 7B.

In each pairwise comparison, Cer-Eval sequentially selects test points, evaluates both models on the selected items, updates its adaptive partition, and continues until the stopping criterion is met. We set the failure probability to $\delta = 0.05$ and report the average saving ratios over 10 replicates in Figure 5. Notably, Cer-Eval consistently outputs correct ranking in all replicates.

Figure 5 demonstrates that **Cer-Eval is particularly advantageous for comparative evaluation.** Unlike static evaluation, which must evaluate all test points, Cer-Eval's sequential design allows early stopping once sufficient statistical evidence is obtained, saving substantial evaluation cost without sacrificing correctness. Cer-Eval automatically adapts to the performance difference between two models: when one model significantly outperforms another, Cer-Eval detects this difference early and terminates the evaluation, achieving saving ratios as high as 98% while maintaining a 95% confidence level. These results highlight Cer-Eval's potential as an efficient, statistically sound framework for large-scale model benchmarking and leaderboard construction, where pairwise or multi-model comparisons are frequent and costly under standard evaluation schemes.

### 6.5 BREAK-EVEN ANALYSIS AND TAKE-AWAYS.

**Break-even Analysis** Cer-Eval offers statistically sound and adaptive evaluation of model performance, with provable cost savings over Static Evaluation under certain conditions. Specifically, Cer-Eval guarantees better efficiency (i.e., evaluates fewer points) than Static Evaluation when $\epsilon \geq \sqrt{[\ln(1/\delta) + \ln \ln(n)]/(2n)}$. For practical settings with $\delta = 0.05$, this condition corresponds to

$\epsilon > 0.014$ on MMLU, $\epsilon > 0.023$ on MATH, and $\epsilon > 0.055$ on AlpacaEval, as shown in Figure 3. Below this threshold, the sequential nature of Cer-Eval may introduce an additional $\sqrt{\ln\ln(n)/(2n)}$ penalty on the estimation error, though in datasets with a benign partition structure (e.g., difficulty groups in MATH) Cer-Eval can still outperform Static Evaluation. A similar break-even phenomenon appears in model comparison, where Cer-Eval becomes advantageous when the expected performance gap between two models exceeds the same threshold.

**Take-away.** Cer-Eval is recommended when the test dataset is large ($\sqrt{[\ln(1/\delta) + \ln\ln(n)]/(2n)} \geq \epsilon$), or when the dataset exhibits a benign partition structure (satisfying Definition 5.1). In these regimes, Cer-Eval provides substantial cost savings while maintaining strict statistical guarantees, making it well suited for large-scale or repeated evaluation tasks.

Nevertheless, the efficiency of Cer-Eval depends on the existence of a benign partition structure. When model performance is nearly uniform across the dataset, as observed in AlpacaEval, the adaptively learned partitions yield limited benefit, and the savings diminish to around 10% even for relaxed error levels. Moreover, in extremely high-accuracy regimes where $\epsilon$ is much smaller than the break-even threshold, Static Evaluation may be preferable due to the unavoidable $\sqrt{\ln\ln(n)/(2n)}$ term introduced by sequential testing.

## 7 CONCLUSION AND FURTHER DISCUSSIONS

In this work, we propose Cer-Eval, an online evaluation framework to assess LLM performance, allowing users to determine their desired evaluation error and confidence level. This approach enables a certified and efficient evaluation process, where users can stop the evaluation once their goal is met, or continue collecting more test points if the current dataset is insufficient for a confident conclusion. Cer-Eval effectively reduces the number of required test points by adapting to each model and dataset of interest. In addition, Cer-Eval is broadly applicable to diverse evaluation tasks, as long as a loss function $\ell$ is defined over input-output pairs.

There are two promising directions for future work. First, extending Cer-Eval to non-IID test scenarios, such as adversarially perturbed or distribution-shifted datasets, presents an important challenge. In such cases, the discrepancy between the test distribution and the intended target distribution can introduce bias in the evaluation results. However, we believe it is possible to quantify and incorporate this bias into the certification guarantees, enabling robust evaluation under more realistic conditions. Second, simultaneous evaluation across multiple tasks presents another opportunity for improvement. Evaluation tasks are known to be often correlated. Exploiting these relationships could further enhance evaluation efficiency.

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

## A MISSING PROOFS

**General notations.** We will use $R$ for short of $R(f, P_{XY})$ when there is no ambiguity. Let $X_i, Y_i, i = 1, \ldots, n$ be IID sampled from $P_{XY}$, and $Z_i = \ell(f(X_i), Y_i)$. Then $Z_i$'s are IID with $0 \leq Z_i \leq 1$, $\mathbb{E}(Z_i) = R(f, P_{XY})$, and $\text{var}(Z_i) = \text{var}(Z_i - 0.5) \leq \mathbb{E}\{(Z_i - 0.5)^2\} \leq 1/4$. Let $\widehat{R}_n = n^{-1} \sum_{i=1}^{n} Z_i$. The online evaluation process will produce a sequence of estimate $\widehat{R}_n$ and confidence interval (CI), characterized by its radius $\epsilon_n$, where $n$ is the number of evaluated test points so far.

**Proof of Theorem 4.2**

*Proof.* We first prove the upper bound by showing that Algorithm 3, denoted as $\mathcal{A}_{\text{seq}}$, is a $(s, \epsilon, \delta)$-certified test algorithm with $s \leq 12\epsilon^{-2}\{\ln(1/\delta) + \ln\ln(1/\epsilon)\}$. Recall that $N$ is the number of test points evaluated when the algorithm terminates. The probability that $\mathcal{A}_{\text{seq}}$ yields a wrong claim is bounded by Lemma B.3, an adaptive Hoeffding-type inequality, as follows:

$$\mathbb{P}(\mathcal{A}_{\text{seq}} \text{ makes a wrong decision}) = \mathbb{P}(\widehat{R}_N - \epsilon_N \geq C) \leq \mathbb{P}(\{\exists n, \widehat{R}_n - R \geq \epsilon_n\}) \leq \delta/2. \quad (1)$$

As for the running time, let $s = \frac{12\ln(4/\delta) + 12\ln\ln(1/\epsilon)}{\epsilon^2}$. We can check that for any sufficiently small $\epsilon$,

$$\epsilon_s^2 = \frac{2\ln(\ln(s) + 1) + \ln(4/\delta)}{s} \leq \epsilon^2/4.$$

Therefore, the probability that the algorithm does not terminate after drawing $s$ samples is

$$\mathbb{P}(N \geq s) \leq \mathbb{P}(\widehat{R}_s + \epsilon_s \geq C) \leq \mathbb{P}(\widehat{R}_s - R \geq \epsilon_s) \leq \delta/2, \quad (2)$$

where the last step is due to Eq. (1). Combining Eq.s (1) and (2) proves that Algorithm 3 is a $(s, \epsilon, \delta)$-certified test algorithm.

Now, we turn to prove the lower bound. Recall that $\epsilon$ can be implicitly determined by $R$, such as in the second estimation goal introduced in Section 3. In particular, we have $\epsilon = |R - C|$ when the user want to determine whether $R$ is above a threshold $C$. In this case, Farrell (1964) proved that for any algorithm that guarantees a $\delta$ failure probability, we have

$$\limsup_{\epsilon \to 0} \mathbb{E}N \geq O\left(\frac{\ln\ln(1/\epsilon)}{\epsilon^2}\right). \quad (3)$$

It is also known that for an easier problem where $\epsilon$ is known in prior (Mannor & Tsitsiklis, 2004), the expected test points for any $\delta$-certified algorithm satisfies

$$\lim_{\epsilon \to 0, \delta \to 0} \mathbb{E}N \geq O\left(\frac{\ln(1/\delta)}{\epsilon^2}\right). \quad (4)$$

Suppose there exists an algorithm that is $(s, \epsilon, \delta)$-certified, where $s$ satisfies

$$\lim_{\epsilon, \delta \to 0} \frac{s\epsilon^2}{\ln(1/\delta) + \ln\ln(1/\epsilon)} = 0. \quad (5)$$

As a result, for any fixed $\epsilon$, there exists a $\delta_0$ such that for any $\delta < \delta_0$, we have

$$\mathbb{P}\left(N > \frac{\ln(1/\delta) + \ln\ln(1/\epsilon)}{\epsilon^2}\right) \leq \delta.$$

Moreover, there exists an integer $n_0 = \frac{\ln(1/\delta_0) + \ln\ln(1/\epsilon)}{\epsilon^2}$ such that for any $n > n_0$, we have

$$\mathbb{P}(N > n) \leq \exp\{-n\epsilon^2 + \ln\ln(1/\epsilon)\}.$$

Then, the following holds for any $s > n_0$:

$$\mathbb{E}N \leq s + \sum_{n=s}^{\infty} \mathbb{P}(N > n) \leq s + \int_s^{\infty} \exp\{-x\epsilon^2 + \ln\ln(1/\epsilon)\}dx \leq s + \frac{\delta}{\epsilon^2}. \quad (6)$$

Finally, comparing Eq. (6) to Eq.s (3) and (4) yields that

$$s \geq O\left(\frac{\ln(1/\delta) + \ln\ln(1/\epsilon)}{\epsilon^2}\right),$$

which contradicts with Eq. (5). We thus completes the proof. $\qquad\square$

**Proof of Theorem 5.2**

*Proof.* First, Theorem 4.2 shows that for $\mathcal{A}_{\text{seq}}$, the needed test sample size $n$ satisfies that

$$n = O\left(\frac{\ln(1/\delta) + 2\ln\ln(1/\epsilon)}{\epsilon^2}\right) \tag{7}$$

We prove that Algorithm 1 is a $(n', \epsilon, \delta)$-certified test algorithm, where $n'$ will be specified below. In particular, the known benign partition will be used as the partition subroutine in the algorithm input. Let $\mu_k := \mathbb{P}(A_i)$ be the probability mass of area $A_k$, for $k = 1, \ldots, K$. Without loss of generality, we assume that $\mu_i$ is known. Otherwise, we can keeping drawing data points (without evaluating them) until the estimation of $\mu_i$ is sufficiently accurate. When a dataset is given instead of the data distribution $P_{X_Y}$, we have $\mu_k = N_k/n$, where $N_k$ is the number of test points in $A_k$.

When Algorithm 1 terminates and evaluate $N$ points, we have

$$\widehat{R}_N - R = \sum_{k=1}^{K} \mu_k(\widehat{R}_k - R_k),$$

where $R_k = \mathbb{E}_{A_k}\ell(f(X), Y)$ is the prediction error on the area $A_k$, and $\widehat{R}_k = n_k^{-1}\sum_{i=1}^{n_k} Z_i^k$ is the empirical loss.

The empirical variance of the loss on $A_k$ is

$$\widehat{v}_k := n_k^{-1}\sum_{i=1}^{n_k}(Z_i^k - \widehat{R}_k)^2 = n_k^{-1}\sum_{i=1}^{n_k}(Z_i^k - R_k)^2 - (R^k - \widehat{R}_k)^2.$$

Let $\eta_k = \sqrt{\{2\ln(\log(n_k) + 1) + \ln(16K/\delta)\}/n_k}$, Lemma B.3 implies

$$\mathbb{P}\left(\left|n_k^{-1}\sum_{i=1}^{n_k}(Z_i^k - R_k)^2 - v_k\right| \le \eta_k\right) \ge 1 - \delta/(4K),$$

$$\mathbb{P}(|R^k - \widehat{R}_k| \le \eta_k) \ge 1 - \delta/(4K).$$

Let $\mathcal{E} = \{v_k \le \widehat{v}_k + \eta_k + \eta_k^2, \widehat{v}_k \le v_k + \eta_k, |R^k - \widehat{R}_k| \le \eta_k\}$, we have $\mathbb{P}(\mathcal{E}) \ge 1 - \delta/2$. Evoking Lemma B.5 on event $\mathcal{E}$, we have

$$\mathbb{P}\left(\widehat{R}_k - R_k \ge \epsilon_k, \mathcal{E}\right) \le \delta/(8K),$$

where

$$\epsilon_k = 2\eta_k^2/3 + 2\sqrt{(\widehat{v}_k + \eta_k + \eta_k^2)\eta_k^2}.$$

As a result, a union bound gives

$$\mathbb{P}(\mathcal{A} \text{ makes a wrong decision}) \le 1 - \mathbb{P}(\mathcal{E}) + \mathbb{P}(\widehat{R}_N - R \ge \sum_k \mu_k\epsilon_k, \mathcal{E})$$

$$\le \delta/2 + \sum_k \mathbb{P}(\exists k, \widehat{R}_k - R_k \ge \epsilon_k) \le 5\delta/8. \tag{8}$$

Regarding the required sample complexity, let $n' = n\ln(K+1)\sum_k \max\{v_k, \epsilon^{2/3}\}$. Suppose the algorithm does not terminate after evaluating $n'$ points. Note that

$$n_i/\{\widehat{v}_i, \eta_i\} = n_j/\max\{\widehat{v}_j, \eta_j\}, 1 \le i \le j \le K, \sum_k n_k = n'. \tag{9}$$

Therefore, $n_k = n\ln(K+1)\frac{\sum_k \max\{v_k, \epsilon^{2/3}\}}{\sum_k \max\{\widehat{v}_k, \eta_k\}}\max\{\widehat{v}_k, \eta_k\}$. Now, we can check that

$$n_k = O(n\ln(K+1)\max\{v_k, \epsilon^{2/3}\}).$$

To see it, when $v_k \geq O(\epsilon^{2/3})$, the above $n_k$ ensures $\eta_k \leq O(\epsilon^{2/3})$, implying that $\max\{\widehat{v}_k, \eta_k\} = \max\{\widehat{v}_k, \epsilon^{2/3}\}$. Similarly, when $v_k \leq O(\epsilon^{2/3})$, $n_k = O(n \ln(K+1)\epsilon^{2/3})$ ensures $\eta_k = O(\epsilon^{2/3})$ and therefore $\max\{\widehat{v}_k, \eta_k\} = \max\{\widehat{v}_k, \epsilon^{2/3}\}$. As a result, this $n_k$ is the solution of Eq. (9).

Finally, on the event $\mathcal{E}$, the above $n'$, up to a universal constant, satisfies that $\epsilon_k \leq \epsilon/2$, implying that $\mathbb{P}(N \geq n', \mathcal{E}) \leq \delta/8$. Thus, Algorithm 1 is a $(n', \epsilon, \delta)$-certified algorithm and we complete the proof. $\qquad\square$

**Proof of Theorem 5.3**

*Proof.* Recall that the loss is bounded in $[0, 1]$ by Theorem 4.1. Given a super-Gaussian loss distribution, the probability mass of $A_k$ is

$$
\mu_k = \mathbb{P}(A_k) \geq A \int_{(k-1)/K}^{k/K} \exp(-z^2/\sigma^2)dz \geq A \int_{(k-1)/K}^{k/K} \exp(-z/\sigma^2)dz
$$

$$
= A\sigma^2 e^{-k/(K\sigma^2)}(e^{1/(K\sigma^2)} - 1) \geq \frac{A}{K}e^{-k/(K\sigma^2)}.
$$

For an equally-spaced partition, we have $v_k \leq 1/K^2$. Therefore, when $K \leq \epsilon^{-1/3}$, Theorem 5.1 is satisfied if

$$
\mu_k \geq \ln(K+1)/K^2, k = 1, \ldots, K
$$

which is equivalent to

$$
K/\ln(K+1) \geq \exp\{1/\sigma^2\}/A. \tag{10}
$$

Therefore, when $K \leq \epsilon^{-1/3}$ and Eq. (10) holds, Theorem 5.2 implies

$$
\rho = O\left( \ln(K+1) \sum_{k=1}^{K} \max\{v_k, \epsilon^{2/3}\} \right) = O(\ln(K+1)/K),
$$

thus completes the proof. $\qquad\square$

# B   TECHNICAL LEMMAS

**Lemma B.1** (Hoeffding Inequality (Boucheron et al., 2013, Theorem 2.8))**.** *For independent observations $X_1, \ldots, X_n$ such that $a_i \leq X_i \leq b_i$ almost surely (a.s.), let $S_n = \sum_{i=1}^{n}\{X_i - \mathbb{E}(X_i)\}$, we have that*

$$
\mathbb{P}(S_n \geq t) \leq \exp\left\{ \frac{-2t^2}{\sum_{i=1}^{n}(b_i - a_i)^2} \right\}.
$$

**Lemma B.2** (Bernstein Inequality (Boucheron et al., 2013, Equation 2.10))**.** *Assume independent observations $X_1, \ldots, X_n$ such that $X_i \leq b$ a.s.. Let $S_n = \sum_{i=1}^{n}\{X_i - \mathbb{E}(X_i)\}$ and $v_n = \sum_{i=1}^{n} \mathrm{var}(X_i)$, we have*

$$
\mathbb{P}(S_n \geq t) \leq \exp\left\{ \frac{-t^2}{2(v_n + bt/3)} \right\},
$$

*or equivalently,*

$$
\mathbb{P}\left( S_n \geq \frac{b\ln(1/\delta)}{3} + \frac{1}{3}\sqrt{b^2\ln^2(1/\delta) + 18v_n\ln(1/\delta)} \right) \leq \delta.
$$

**Lemma B.3** (Adaptive Hoeffding Inequality (Zhao et al., 2016, Theorem 1))**.** *Let $\epsilon_n = \sqrt{\frac{2\ln(\log(n)+1)+\ln(4/\delta)}{n}}$. For independent observations $X_1, \ldots, X_n$ such that $0 \leq X_i \leq 1$ a.s., let $S_n = n^{-1}\sum_{i=1}^{n}\{X_i - \mathbb{E}(X_i)\}$. Then, we have*

$$
\mathbb{P}(\{\exists n, S_n/n \geq \epsilon_n\}) \leq \delta/2.
$$

**Lemma B.4** (Maximal Form of Bernstein Inequality (Kevei & Mason, 2011))**.** *Assume independent observations* $X_1, \ldots, X_n$ *such that* $X_i \leq b$ *a.s.. Let* $S_n = \sum_{i=1}^n \{X_i - \mathbb{E}(X_i)\}$ *and* $v_n = \sum_{i=1}^n \mathrm{var}(X_i)$, *we have*

$$\mathbb{P}(\max_{1 \leq i \leq n} S_i \geq t) \leq \exp\left\{\frac{-t^2}{2(v_n + bt/3)}\right\},$$

*or equivalently,*

$$\mathbb{P}\left(\max_{1 \leq i \leq n} S_i \geq \frac{b \ln(1/\delta)}{3} + \frac{1}{3}\sqrt{b^2 \ln^2(1/\delta) + 18 v_n \ln(1/\delta)}\right) \leq \delta.$$

**Lemma B.5** (Adaptive Bernstein Inequality )**.** *Let* $u_n = 2\ln(\log(n) + 1) + \ln(4/\delta)$ *and* $\epsilon_n = (bu_n + \sqrt{b^2 u_n^2 + 18 v_{2n} u_n})/(3n)$. *For independent observations* $X_1, \ldots, X_n$ *such that* $0 \leq X_i \leq 1$ *a.s., let* $S_n = \sum_{i=1}^n \{X_i - \mathbb{E}(X_i)\}$. *Then, we have*

$$\mathbb{P}(\{\exists n, S_n/n \geq \epsilon_n\}) \leq \delta/2.$$

*Proof.* Applying Lemma B.4 yields

$$
\begin{aligned}
\mathbb{P}(\{\exists n, S_n/n \geq \epsilon_n\}) &= \mathbb{P}(\cup_{n=1}^\infty \{S_n \geq n\epsilon_n\}) \\
&= \mathbb{P}(\cup_{l=0}^\infty \cup_{2^l \leq n \leq 2^{l+1}} \{S_n \geq n\epsilon_n\}) \\
&\leq \mathbb{P}(\cup_{l=0}^\infty \{\max_{2^l \leq n \leq 2^{l+1}} S_n \geq 2^l \epsilon_{2^l}\}) \\
&\leq \sum_{l=0}^\infty \mathbb{P}\left[\max_{1 \leq n \leq 2^{l+1}} S_n \geq 2^l \epsilon_{2^l}\right] \\
&\leq \sum_{l=0}^\infty e^{-u_{2^l}} = \sum_{l=0}^\infty (l+1)^{-2} \delta/4 \leq \delta/2,
\end{aligned}
$$

thus completes the proof. $\square$

## C    MISSING ALGORITHMS

---
**Algorithm 3** Vanilla Online Evaluation (Seq)

---
**Require:** The estimation error level $\epsilon$, failure probability $\delta$, and test dataset $D$
 1: Random shuffle $D$
 2: **for** Round $n = 1, 2, \ldots$ **do**
 3:     Let $\epsilon_n \leftarrow \sqrt{\frac{2\ln(\log(n)+1)+\ln(4/\delta)}{n}}$
 4:     Evaluate the loss $Z_n \leftarrow \ell(f(X_n), Y_n)$
 5:     Let $\widehat{R}_n \leftarrow n^{-1} \sum_{i=1}^n Z_i$
 6:     **if** $\epsilon_n \leq \epsilon$ **then**
 7:         Terminate and return $\widehat{R}_n, \epsilon_n$
 8:     **end if**
 9: **end for**
**Output:** The estimated loss $\widehat{R}_n$, confidence interval radius $\epsilon_n$, and number of evaluated points $n$

---

## D    EXPERIMENT DETAILS AND FURTHER EXPERIMENTS

**Compute Resources**

All experiments are conducted on a computing cluster equipped with 8 NVIDIA A100 GPUs, each with 80 GB of HBM2e memory. While the GPUs are primarily used to serve the evaluated models, our algorithm is lightweight and can be efficiently run on CPUs with minimal memory requirements.

**Models.** We categorize the HuggingFace links or endpoint specification of all the models used in the evaluation as follows.

Table 3: HuggingFace links or endpoint specifications of evaluated models.

| Model | Endpoint |
|---|---|
| GPT-4o | `https://platform.openai.com/docs/models/gpt-4o`, `gpt-4o-2024-11-20` endpoint |
| Llama3-8B | `https://huggingface.co/meta-llama/Meta-Llama-3-8B` |
| Mistral-7B | `https://huggingface.co/mistralai/Mistral-7B-v0.3` |
| Qwen2-7B | `https://huggingface.co/Qwen/Qwen2-7B` |

Table 4: Saving ratio $\rho$ by 'Subset' and Cer-Eval for various $\epsilon$ values (Scenario 2).

| | $\epsilon$ | 0.0177 | 0.0210 | 0.0255 | 0.0310 | 0.0377 | 0.0458 | 0.0557 | 0.0677 | 0.0821 | 0.0998 |
|---|---|---|---|---|---|---|---|---|---|---|---|
| Subset | $\rho$ | 0.048 | 0.322 | 0.541 | 0.689 | 0.789 | 0.857 | 0.903 | 0.934 | 0.955 | 0.969 |
| Cer-Eval | $\rho$ | 0.0004 | 0.611 | 0.719 | 0.794 | 0.849 | 0.888 | 0.916 | 0.937 | 0.954 | 0.969 |

**Comparison with evaluating a randomly sampled subset.** We compare Cer-Eval to a new baseline, Subset, which randomly samples $\eta$ proportion of the test points and reports the average score and error level. The saving ratio is defined as $\rho = 1 - \eta$. Below, we present results comparing Subset and Cer-Eval under the same simulation settings as our Scenarios 2 (easy-to-distinguish) and 3 (hard-to-distinguish). We summarize how the achieved error level $\epsilon$ corresponds to the savings ratio $\rho$ in Tables 4 and 5.

Cer-Eval outperforms the Subset baseline in Scenario 2 due to its ability to adaptively explore the variance structure of the dataset and model. However, in Scenario 3, where a benign partition is difficult to identify, Cer-Eval may need to evaluate more points than Subset, a cost for providing an any-time valid guarantee.

**Ablation study for the influence on embedding models**

We consider three embedding models: a pre-trained BERT, or text-embedding-3-large and text-embedding-ada-002 model from OpenAI. With the same setting as the real-world experiments performed in Section 6, we report the saving ratios for evaluating four models on three datasets in Figures 6, 7 and 8. Also, the empirical failure probability is zero for all settings. We find that using different embedding models leads to highly similar results. Nevertheless, it does not imply that embedding model is unimportant. In contrary, the efficiency of Cer-Eval heavily depends on the goodness of partition. This is confirmed by our simulation studies, where Cer-Eval performs better in the easy-to-distinguish scenario. For these datasets, we conjecture that the proposed partition method, Algorithm 2, does not extract enough information from the embedding vectors. We anticipate a higher saving ratio if more informative embeddings of the queries are available, or a more effective partition algorithm is deployed. One potential direction is to use the intrinsic attributes of queries as embedding vectors, such as the one used in (Polo et al., 2025).

**Test scaling law.** We further evaluated over 200 models on the AlpacaEval dataset to investigate potential factors affecting test sample complexity, analogous to the training scaling law (Kaplan et al., 2020; Hoffmann et al., 2022; Bahri et al., 2024). For multiple model families, Figure 9 shows the relationship between model size and the needed test sample size for a certifiable evaluation within error level $\epsilon = 0.06$ when using Cer-Eval. At first glance, it seems a larger model requires more test points, resembling the training scaling law. However, Figure 10 shows that the this trend is spurious: larger models require more test points on the AlpacaEval dataset because their accuracy is closer to 0.5, leading to higher loss variance. We also plot the accuracy v.s. test sample size curve for over 200 models on MATH dataset, as shown in Figure 11.

This finding suggests an intriguing connection between model performance and test sample complexity, offering insights into leveraging the training scaling law. Suppose an LLM continues to scale and achieves higher accuracy (above 0.5), fewer test points are sufficient for evaluation. In other words,

Table 5: Saving ratio $\rho$ by 'Subset' and Cer-Eval for various $\epsilon$ values (Scenario 3).

| | $\epsilon$ | 0.0177 | 0.0210 | 0.0255 | 0.0310 | 0.0377 | 0.0458 | 0.0557 | 0.0677 | 0.0821 | 0.0998 |
|---|---|---|---|---|---|---|---|---|---|---|---|
| Subset | $\rho$ | 0.048 | 0.322 | 0.541 | 0.689 | 0.789 | 0.857 | 0.903 | 0.934 | 0.955 | 0.969 |
| Cer-Eval | $\rho$ | 0.0006 | 0.28 | 0.510 | 0.665 | 0.770 | 0.843 | 0.890 | 0.926 | 0.946 | 0.963 |

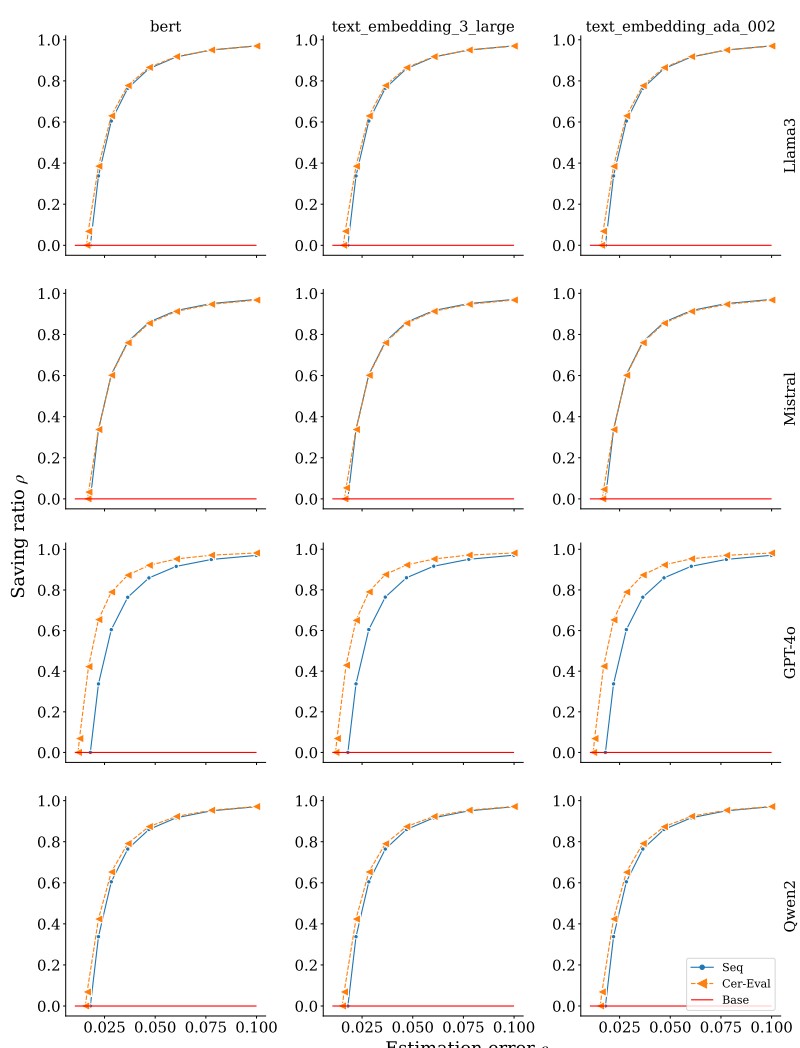

Figure 6: Percentage of test points saved by the proposed algorithms compared to Base when evaluating models on the MMLU dataset.

despite increasing model sizes, we may not need a growing test dataset for LLM evaluation with a fixed estimation error level and confidence level. It therefore implies an encouraging prospect for future LLM development and evaluation.

**Choice of partition numbers and partition subroutines** A key advantage of Cer-Eval is that the partition number $K$ can be automatically determined by Algorithm 2, eliminating the need for manual selection or fine-tuning. For example, our real-world experiments used adaptive partition without fine-tuning $K$. However, users can opt for a fixed partition with a user-specified $K$. In this case, our simulations show that an improper $K$ may reduce evaluation efficiency, requiring more test points to maintain the same error and confidence level. Thus, we recommend practitioners to run Algorithm 1 with the adaptive partition subroutine by default.

As for partition subroutines, technically speaking, any classification method can be used, though partition quality affects evaluation efficiency. We chose $k$-NN for its simplicity and intuitive appeal – test points close in the representation space should yield similar performance. We tried $k$-NN with various values of $k$, including $k = \ln(n)$ and $k = n^{-\alpha}$ for some $\alpha > 0$. Compared to 1-NN, we found that these choices degraded evaluation efficiency, especially on imbalanced datasets. This is likely because $k$-NN is susceptible to imbalanced data: a large $k$ may lead to a biased classifier

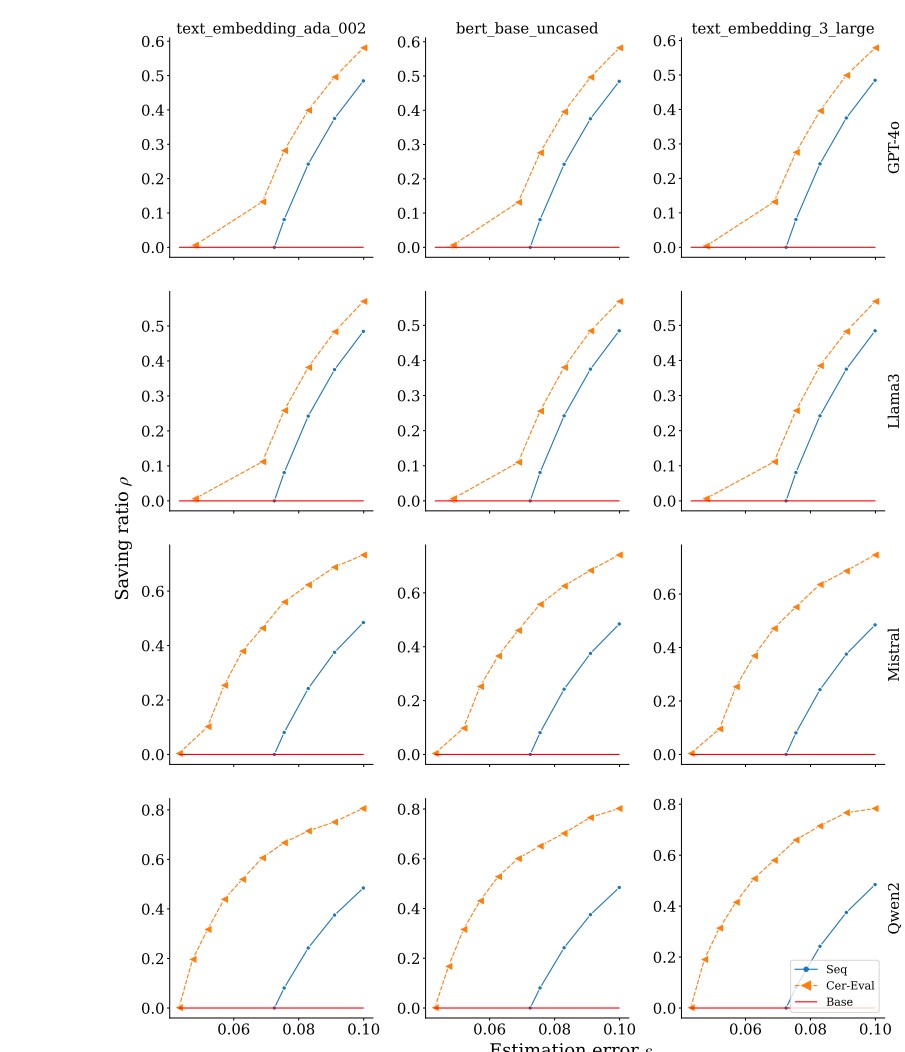

Figure 7: Percentage of test points saved by the proposed algorithms compared to Base when evaluating models on the AlpacaEval dataset.

that favors the majority class. We thus chose $k = 1$ in our experiments to mitigate bias and reduce computational complexity.

In addition, the partition can be done on other representation of data, not only text embedding. Representation quality affects partition effectiveness and therefore evaluation efficiency. We chose text embeddings for clarity and accessibility, as they are available from open-source models. A promising future direction is to explore alternative features that better capture performance complexity.

# E    IMPACT STATEMENT

Our work enables a certifiable and efficient assessment of various aspects of LLMs, including their capabilities, robustness to adversarial prompts, and alignment with human values. We therefore anticipate a positive societal impact, as timely and rigorous evaluation plays a crucial role in enhancing AI responsibility, mitigating potential risks, and ensuring that AI technologies align with ethical and safety standards.

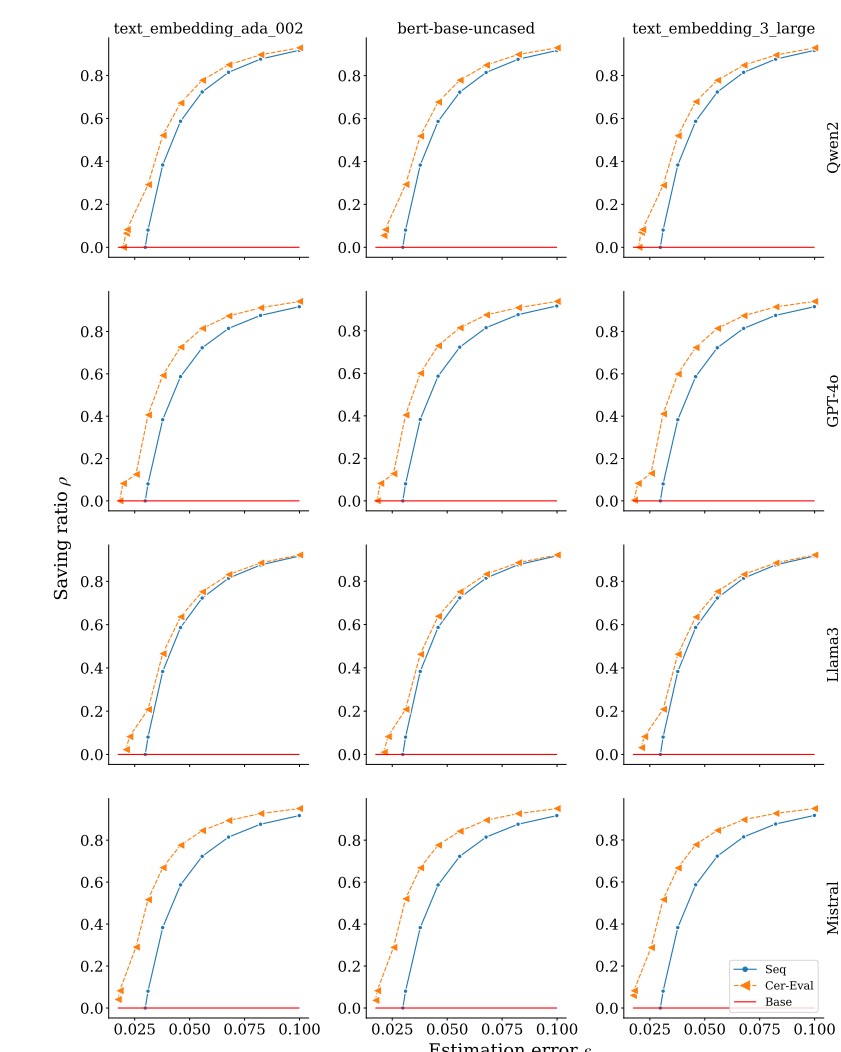

Figure 8: Percentage of test points saved by the proposed algorithms compared to Base when evaluating models on the MATH dataset.

## F    THE USE OF LARGE LANGUAGE MODELS STATEMENT

Large language models were used solely as a writing aid. Their use was limited to minor language editing, such as correcting grammar, improving clarity, and polishing the phrasing, without altering the substantive content or analysis of the article.

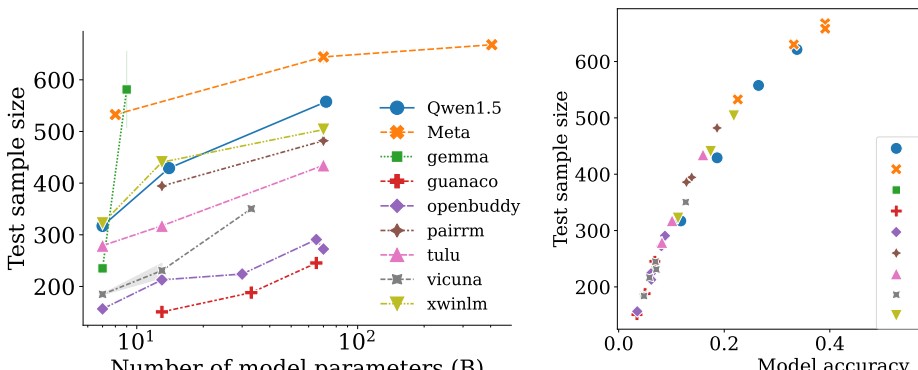

Figure 9: Number of needed test points v.s. model size when evaluating models from multiple families using Cer-Eval, with estimation error level $\epsilon = 0.07$ and failure probability $\delta = 0.05$.

Figure 10: Number of needed test points v.s. model accuracy when evaluating models from multiple families using Cer-Eval, with estimation error level $\epsilon = 0.07$ and failure probability $\delta = 0.05$.

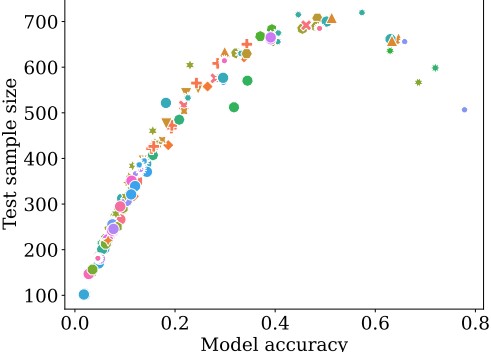

Figure 11: Number of needed test points v.s. model accuracy when evaluating models from multiple families using Cer-Eval, with estimation error level $\epsilon = 0.07$ and failure probability $\delta = 0.05$.

