# OpenReview forum: "Cer-Eval: Certifiable and Cost-Efficient Evaluation Framework for LLMs"
_ICLR.cc/2026/Conference — Submitted to ICLR 2026_

### Official Review · Reviewer_xqW7 · 2025-10-25

**Soundness:** 4
**Presentation:** 3
**Contribution:** 3
**Rating:** 6
**Confidence:** 3

**Summary:**

The authors argue that existing evaluations in LLMs do not (i) adapt to specific user goals, and (ii) do not consider trade-offs between evaluation cost and validity making it difficult to choose the correct amount of evaluation data. TO address this, the authors introduce Cer-Eval, an online evaluation framework for large language models that provides statistical guarantees while reducing evaluation costs. They introduce the concept of "sample complexity" to define the minimum number of test points required to obtain a reliable evaluation result. They provide certified bounds on bounded loss functions on the basis of sample complexity and empirically show that they are able to reduce test samples on common benchmarks considerably while maintaining evaluation accuracy.

**Strengths:**

- Addresses two underexplored but important problems: (i) certifiable / reliable, and (ii) efficient evaluation of LLMs.
- The authors provide theoretical motivation and formal guarantees for their framework
- Empirical evidence is provided to show that this method can lead to practical efficiency gains on popular benchmarks
- The authors provide relevant context regarding prior work in this area, specifically work concerned with efficient LLM evaluations
- Ablation on the influence of the embedding model is provided in the appendix. I would consider moving this analysis to the main paper (or at least a fraction of it)

**Weaknesses:**

- The experimental validation focuses mainly on accuracy and efficiency. I would have liked an evaluation of the robustness of the confidence intervals (e.g., with minor perturbations to the data or repeated experiments)
- It should be emphasized, that these results only hold if the partition is i.i.d. with respect to the rest of the evaluation set, which might not always be true in practice (or under adversarial attacks / poisoning / etc.)
- No evaluation of the overhead of the algorithm, despite the focus on efficiency. What is the total run-time of evaluations? (Efficiency gains are sometimes around 20% without considering the overhead)

**Questions:**

- Would it be possible to adversarial attacks this framework? E.g., could an attacker design the data samples in a manner where uninformative subsets are chosen for evaluation? After all, the framework uses an embedding function, which will likely be vulberable to adversarial attacks
- Did the authors consider ranking different models based on metrics derived from their evaluations? If yes, are these rankings consistent with full evaluations?
- What is the computational overhead of adaptive partitioning compared to static evaluation? Benchmarks, such as MMLU can be evaluated on a single H100 GPU in ~20 Minutes on a 8B parameter model. What is the overhead of the adaptive partitioning?
- Could the guarantees be extended to non-i.i.d. settings? (I dont see this as a weakness of the paper just curious)
- Similarly, how would this approach scale to millions of data samples, which is the main use-case of this framework?
- Would the framework also be applicable to binary metrics / other language modeling tasks, such as classification?  Based on the provided theory I would assume so, the authors could discuss this in the outlook of their work.

I am strongly considering to raise my score based on the authors feedback and the other reviews.

---

> ### Author Response · Authors · 2025-11-18
>
> We sincerely thank the reviewer for recognizing the significance of our work and for the constructive feedback. In response, we have revised the manuscript to incorporate your suggestions. Major changes are highlighted in blue. Below, we summarize our key revisions and clarifications:
>
> **Vulnerability to adversarial attacks. (Section 7)** Yes, it is possible to adversarially attack Cer-Eval. For instance, when Cer-Eval performs adaptive partitioning using embeddings, adversarially chosen test points can distort the local variance structure, thereby reducing the effectiveness of the partitioning and diminishing efficiency. Moreover, such attacks effectively shift the test distribution away from the original one, introducing potential bias in the evaluation outcome.
>
>
> **Comparing multiple models. (Section 6.4)**
>
> While Cer-Eval is proposed to evaluate a single model, it naturally extends to broader applications such as pairwise model comparison and ranking multiple models. As demonstrated in Section 6.4, we applied Cer-Eval to pairwise model comparison on the MMLU benchmark. We consider four models with the ground-truth ranking as GPT-4o > Qwen2 7B > LLaMA3 8B > Mistral 7B.
>
> Our experimients demonstrates that **Cer-Eval is particularly advantageous for comparative evaluation.** Unlike static evaluation, which must evaluate all test points, Cer-Eval’s sequential design allows early stopping once sufficient statistical evidence is obtained, saving substantial evaluation cost without sacrificing correctness. Cer-Eval automatically adapts to the performance difference between two models: when one model significantly outperforms another, Cer-Eval detects this difference early and terminates the evaluation, achieving saving ratios as high as 98% while maintaining a 95% confidence level. These results highlight Cer-Eval’s potential as an efficient, statistically sound framework for large-scale model benchmarking and leaderboard construction, where pairwise or multi-model comparisons are frequent and costly under standard evaluation schemes.
>
> **Computational overhead. (Section 6.2, Lines 453-462)**
>
> The computational overhead of Cer-Eval primarily stems from the adaptive partitioning step. However, in practice, this step is lightweight. For example, on the MMLU dataset evaluated with an 8B parameter model, the adaptive partitioning takes only a few seconds on a single H100 GPU, while full model inference takes ~20 minutes.
>
> For large-scale datasets, Cer-Eval can further reduce this cost through scheduled partition updates, instead of re-partitioning after each test point. For example, we can perform partitioning after evaluating $\alpha \beta^t $ points ($\alpha>0, \beta>1$, and $t=1,2,\dots$) or after every fixed percentage of the dataset (e.g., $1\%$). Under such schedules, the partition step is executed at most a fixed number of times $T$. As each partitioning step using the $1$-NN algorithm has time complexity $O(n\ln(n))$, the overall time complexity of partitioning is $O(Tn\ln(n))$, which is almost linear in the dataset size. Thus, Cer-Eval scales efficiently to large test sets.
>
> **Non-IID setting. (Remark 5.7)**
>
> We acknowledge that Cer-Eval assumes i.i.d. test samples from the target distribution. When this assumption is violated, evaluation results may be biased. A possible extension is to model the distribution shift between the target $P$ and the actual test distribution $Q$. In this case, the $(\epsilon,\delta)$-evaluation-guarantee of Cer-Eval can be adjusted to $(\epsilon+dist(P,Q),\delta)$-guarantee, where $dist(P,Q)$ measures the divergence between the two distributions. Formalizing this extension is an exciting direction for future work.
>
>
> **Application to general evaluation tasks. (Section 7)**
>
> Yes, Cer-Eval can be applied to a wide range of evaluation tasks, including binary and multi-class classification. As long as a loss function $\ell$ is defined over input-output pairs, Cer-Eval can estimate the expected loss $R(f, P_{XY})= E_{X,Y}\ell(f(X),Y)$. For example, in binary classification, one may use the 0-1 loss or cross-entropy loss, and Cer-Eval remains applicable without modification.
>
>
> We hope our revisions and new results adequately address your concerns, and we are happy to clarify or expand on any other questions you may have.

---

> > ### Comment · Reviewer_xqW7 · 2025-11-19
> > **Thank you for addressing my concerns**
> >
> > I have read the other reviews and rebuttals, and I am overall satisfied with the author's responses. Specifically regarding limited comparisons to similar methods proposed in earlier work (Reviewer mNqD), and I personally do not agree that there is limited discussion on "real-world" settings.
> >
> > My own concerns were also addressed.
> >
> > **Vulnerability and non-iid**
> > Thank you for providing additional information regarding limitations related to data poisoning or non-i.i.d. settings (which should be briefly mentioned in a limitations section at the end of the work). However, I acknowledge that these settings are out of scope of this work and don't view these limitations as a weakness of the work (I just think they should be shortly discussed).
> >
> > **Compute overhead**
> > I think adding this information to the paper would strengthen the work.
> >
> > **Model comparison and application to other tasks**
> > Thank you for clarifying my misunderstandings regarding model comparison and application to other tasks. As other reviewers had similar concerns, I would suggest extending explanations in this direction in the camera-ready version.
> >
> > I recommend accepting this paper to the conference as it addresses a common problem in LLMs (reliable evaluation) in a practical and principled manner. I changed my score accordingly.

---

> > > ### Author Response · Authors · 2025-11-19
> > >
> > > Thank you very much for taking time to carefully read our responses to all reviewers. We are very pleased to hear that our rebuttals have satisfactorily addressed your concerns, and that you found our clarifications helpful in the context of other reviewers’ comments as well.
> > >
> > > We sincerely appreciate your decision to raise the score and your recognition that our work addresses a critical issue in LLM evaluation in both a practical and principled manner. We are committed to incorporating all the discussions into the revised manuscript.
> > >
> > > Thank you again for your thoughtful review and support!

---

### Official Review · Reviewer_MFBd · 2025-10-26

**Soundness:** 2
**Presentation:** 3
**Contribution:** 2
**Rating:** 4
**Confidence:** 4

**Summary:**

This paper introduces a principled, online evaluation framework for LLM evaluations. The framework is based on partitioning the embedding space, and improves efficiency by leveraging variance of the evaluation in each partition. It can provide evaluations with high-confidence while requiring only 60-90% of the training data in empirical evaluations on benchmark datasets.

**Strengths:**

- The proposed evaluation approach seems to reduce the number of test samples required to obtain a confident evaluation, which can be of great use to practitioners since LLM evaluations are typically computationally expensive.
- The proposed evaluation is principled and theoretically motivated.
- The paper provides extensive evaluations on a synthetic task and for four models on three "real-world" datasets, empirically validating the effectiveness of their proposed evaluation.

**Weaknesses:**

*Missing breakeven analysis.* The discussion in the paragraph starting line 888 is not sufficient. The paper would require a thorough discussion on the breakeven point, i.e. the point at which using this evaluation is better than evaluating on the entire dataset. I am not sufficiently convinced that the proposed approach is always preferable - as the user has to tolerate a moderate to larger estimation error and/or the test dataset needs to be sufficiently large. I think the point where your approach becomes beneficial should be identified with more certainty, and would be required for practitioners to decide when implementing your approach provides a benefit. A more careful analysis and discussion would be critical for this paper and this should be, in particular, discussed in the main text.

*Limited discussion of real-world experiments.* The discussion on the evaluation on "real-world" datasets is quite limited. While sufficiently many models are tested, the paper would benefit from a more comprehensive discussion of the results, and in particular of the limitations of this approach. The current discussion of the empirical results in real-world settings fails to offer clear takeaways.

*Overly complicated formalization.* The formalization could be improved to make the paper more readable. For example, at first $n$ is introduced as the dataset size, but in Def. 3.6 $n$ represents a function depending on several parameters and this dependency remains unclear. The current formalization is not clear enough, especially not helpful for practitioners who just need to implement your approach. It becomes especially confusing in Theorem 4.2 where the function is denoted as $n'$. Generally all the variables $n, n', n^*$ and $N$ require disambiguation.

Overall, I think the paper has potential toward improved LLM evaluations but the mentioned weaknesses should be addressed to improve clarity and to provide more clear takeaways in particular for practitioners.

**Questions:**

Why do you focus on MMLU, AlpacaEval and MATH in particular? Are these datasets suited to properly evaluate your proposed evaluation approach in more realistic settings? Could additional datasets be helpful to provide more objective insights?

---

> ### Author Response · Authors · 2025-11-18
>
> We sincerely thank the reviewer for the constructive feedback. In response, we have revised the manuscript to address your comments, with major changes highlighted in blue. Below, we summarize our key revisions.
>
> **Breakeven analysis and discussion of experiments.**
>
> As suggested, we have rewritten the breakeven analysis and included it in the main paper (Section 6.5). We have clearly articulated the breakeven condition under which Cer-Eval becomes cost-effective. Specifically, **Cer-Eval is guaranteed to reduce evaluation cost compared to Static Evaluation when** $\sqrt{[\ln(1/\delta)+\ln\ln(n)]/(2n)} \geq \epsilon$, where $n$ is the dataset size, $\epsilon$ the desired estimation error level, and $\delta$ the failure probability. When this condition does not hold, the sequential nature of Cer-Eval introduces an additional error term of order $\sqrt{\ln\ln(n)}/(2n)$.
>
> We also provide a revised discussion summarizing **both the advantages and limitations** of Cer-Eval:
> - Cer-Eval is beneficial when dataset is relatively large compared to the desired estimation error level ($\sqrt{[\ln(1/\delta)+\ln\ln(n)]/(2n)} \geq \epsilon$), or when dataset exhibits a benign partition structure (satisfying Definition 5.1).
> - Nevertheless, the efficiency of Cer-Eval depends on the existence of a benign partition structure. When model performance is nearly uniform across the dataset, as observed in AlpacaEval, the adaptively learned partitions yield limited benefit. Moreover, in extremely high-accuracy regimes where $\epsilon$ is much smaller than the break-even threshold, Static Evaluation may be preferable due to the unavoidable $\sqrt{\ln\ln(n)}/(2n)$ term introduced by sequential testing.
>
> **Clarification of Notation.**
>
> To eliminate the ambiguity of $n$, we have revised Definition 3.4 and Theorem 4.2. In particular, we now use the term ''$(s,\epsilon,\delta)$-certified'' to characterize the guarantees provided by an evaluation algorithm, where $s$ denotes the number of required sample size. This avoids confusion with the dataset size and improves clarity.

---

### Official Review · Reviewer_mNqD · 2025-10-31

**Soundness:** 2
**Presentation:** 4
**Contribution:** 3
**Rating:** 4
**Confidence:** 4

**Summary:**

This paper looks into the problem of efficient and reliable evaluation of LLMs. The authors propose a method called Cer-Eval, an online evaluation framework that sequentially select test points until a user specified estimation error and confidence level is achieved. The paper defines the concept of "test sample complexity" by the minimum number of test samples required for the "certified evaluation", and introduces theoretical formulation for the problem. Based on the definition, the paper derives upper and lower bounds based on the test sample complexity. The authors then developing an empirical partition-based adaptive algorithm that exploits variance structure within each partition to reduce evaluation cost. The experiment results on a few real world benchmarks show 20-40% test samples saving with the same estimation accuracy and 95% confidence interval against baselines.

**Strengths:**

1. The problem is well defined and the formulation is well motivated. The paper addresses a genuinely critical problem. The problem is well-known, yet many papers (both from industry and academic) are still using static evaluation which is sample inefficient.

2. The paper lays a strong theoretical foundation and introduces principled approach to variance reduction. The authors have done a great analysis matching the upper and lower bounds on the test sample complexity, and the theorem 5.2 provides clear connections between partition quality and sample complexity.

3. In addition to the theoretical analysis, the paper also proposes a practical algorithm to apply the method to realistic scenarios where the true partitions of the test data is not available. The experiments results show that the designed procedures are able to save the evaluation cost by 20-40%.

**Weaknesses:**

1. My biggest concern for the paper is the critical gap in baseline comparisons. The paper only effectively compare the method with two baselines, i.e., the static evaluation and vanilla online evaluation process. There are quite a few obvious papers that address the same problem are not included. For example, TinyBenchmarks (Polo et all., 2024) leverages Item Responses Theory (IRT) and is able to achieve <2% estimation error based on 1% of the full MMLU dataset. Similarly, StratPPI (Fisch et al., 2024) and AutoEval (Boyeau et al., 2024) is cited in the related works but are not compared as baseline. Although some of the work requires the partition to be known in advance, however, I don't think that itself can justify the absence of these baselines. I would very strongly suggest the authors to including more baselines for comparison.

2. My another concern is the experimental setup. The main focus of the paper focuses on the efficiency and reliability of the evaluations, and the experiments solely base on the saving ratio vs. estimation error level. However, in practice, the most important goal for evaluation is for model comparisons (i.e., model A vs. model B). Why we don't carry on experiments to show the effectiveness of the method on ranking models? For example, are we able to rank models consistently with the 40% saving on the computation cost? I really think these experiments are critical to show the effectiveness of the proposed method and push the adoption.

**Questions:**

1. Why we are not comparing our method with other baselines? Any specific reason we only pick these two baselines in the paper?
2. Have we done experiments to show how the proposed method rank models again baselines (e.g., static evaluation)? Since the method based on initial partition, how does the initial partition impact the results and how sensitive is our method to the initial partition?
3. Have we done any experiment to understand how the Cer-Eval handles non-deterministic LLMs (e.g., using large temperature etc.) there the f(X) varies across runs? Is the results sensitive to that?

---

> ### Author Response · Authors · 2025-11-18
>
> We sincerely thank the reviewer for their thoughtful feedback, especially regarding our experimental design. To address your concerns, we conducted additional experiments covering: (1) comparisons with more baseline methods, and (2) evaluations involving multiple models.
>
> These new results and discussions have been incorporated into the revised manuscript (Sections 6.3 and 6.4, highlighted in blue). Below, we summarize our key responses:
>
> **Comparison with Polo et al. and Fisch et al.**
>
> We compared Cer-Eval against p-IRT and gp-IRT, two IRT-based methods proposed by Polo et al. (2025), on the MMLU and AlpacaEval benchmarks using the authors’ released code. In both cases, p-IRT and gp-IRT evaluated 100 pre-selected points, while Cer-Eval stopped after evaluating 100 adaptively chosen points.
>
> **The results indicate that p-IRT and gp-IRT do not significantly outperform Cer-Eval in terms of estimation error. However, Cer-Eval uniquely provides formal statistical guarantees on the evaluation accuracy, which IRT-based methods lack.** For instance, with 100 evaluated points, Cer-Eval ensures that the estimated score deviates by at most 0.12 with 95% confidence and 0.08 with 75% confidence. By contrast, IRT-based methods offer no certified error bounds, leaving practitioners uncertain about how accurate the evaluation is or whether 100 points suffice.
>
> We next compare Cer-Eval with with StratPPI, a stratification-based method from Fisch et al. StratPPI assumes a known partition of the dataset and evaluates \textit{all} test points. When such a partition is available, StratPPI achieves slightly better accuracy. However, in practical scenarios where partitions are not available, Cer-Eval adaptively constructs effective partitions and can outperforms StratPPI. Moreover, StratPPI must evaluate every test point, while Cer-Eval can achieve a moderate desired error level (e.g., $\epsilon > 0.02$, $\delta = 0.05$) with far fewer queries .
>
> **Comparing multiple models.**
>
> While Cer-Eval is proposed to evaluate a single model, it naturally extends to broader applications such as pairwise model comparison and ranking multiple models. To illustrate this potential, we conduct an empirical demonstration of pairwise model comparison on the MMLU benchmark. We consider four models with the ground-truth ranking as GPT-4o > Qwen2 7B > LLaMA3 8B > Mistral 7B.
>
> Our experimients demonstrates that **Cer-Eval is particularly advantageous for comparative evaluation.** Unlike static evaluation, which must evaluate all test points, Cer-Eval’s sequential design allows early stopping once sufficient statistical evidence is obtained, saving substantial evaluation cost without sacrificing correctness. Cer-Eval automatically adapts to the performance difference between two models: when one model significantly outperforms another, Cer-Eval detects this difference early and terminates the evaluation, achieving saving ratios as high as 98% while maintaining a 95% confidence level. These results highlight Cer-Eval’s potential as an efficient, statistically sound framework for large-scale model benchmarking and leaderboard construction, where pairwise or multi-model comparisons are frequent and costly under standard evaluation schemes.
>
> **Non-deterministic LLMs.** Cer-Eval accommodates non-deterministic models. When a random function $f(X)$ is used, the definition of prediction error is generalized to $R(f, P_{XY})= E_{f,X,Y}\ell(f(X),Y)$, which includes the variation from both the data and model randomness. Importantly, **the procedure and guarantee of Cer-Eval remains unchanged**, as Cer-Eval focuses on certifiable and efficient estimation of the quantity $R(f, P_{XY})$, thus is irrelavent to the form of $f$.
>
> In the scenario where $f$'s randomness is increased, e.g., by using a larger temperature, $f$'s' variance is increased, leading to less benign partitions and therefore undermining the efficiency of Cer-Eval. Nonetheless, even in such high-variance settings, Cer-Eval still offers valid statistical guarantees and can save cost compared to static evaluation when the desired error level is moderate.
>
>
> We hope our additional experiments have addressed your concerns, and we’d be happy to clarify or expand on any other questions you may have.

---

### Author Response · Authors · 2025-11-29
**Revision Summary**

Dear Reviewers and Area Chair,

We sincerely appreciate your constructive feedback and the time you’ve dedicated to the review process. We especially appreciate Reviewer xqW7’s follow-up, in which they **recommended acceptance (increased score from 6 to 8, post-discussion/pre-leakage)** and **noted that our responses satisfactorily address the concerns raised by the other reviewers**. Below, we summarize the major revisions and clarifications we have made in response to your comments. These updates are highlighted in blue in the revised manuscript.

1. **Experimental Comparison with Existing Work.** We have incorporated comparisons to Polo et al. and Fisch et al., as suggested by Reviewer mNqD. The results show that (1) Polo et al. do not significantly outperform Cer-Eval, and unlike Cer-Eval, they lack formal statistical guarantees; and (2) StratPPI achieves slightly better accuracy when an optimal partition is known in advance, whereas Cer-Eval can achieve a moderate error level with substantially fewer queries.

2. **Model Comparison Experiments.** In response to Reviewers mNqD and xqW7, we added pairwise model comparison experiments on the MMLU benchmark to demonstrate that Cer-Eval naturally extends to tasks like model comparison and ranking. These experiments show that Cer-Eval's sequential design allows early stopping once sufficient statistical evidence is gathered, making it especially effective for comparative evaluation.

3. **Refined Breakeven Analysis.** As suggested by Reviewer MFBd, we have now clearly articulated the breakeven condition under which Cer-Eval becomes cost-effective in Sectino 6.5. We also provide a revised discussion summarizing both the advantages and limitations of Cer-Eval.

4. **Limitations in Non-i.i.d. Settings.** Following Reviewer xqW7’s comments, we now discuss potential vulnerabilities of Cer-Eval under data poisoning or non-i.i.d. conditions.


We believe these revisions comprehensively address the reviewers’ concerns. We kindly hope the Area Chair will take these updates into account during the final evaluation. Thank you again for your valuable time!

Sincerely,

Authors of Submission 13327

---

### Meta-Review · Area_Chair_pETX · 2026-01-08

**Summary:**

This paper receives 3 high-quality reviews, with 1 positive rating (6) and 2 negative ratings (4, 4).

The positive reviewer previously has some concerns, which have been addressed during the rebuttal.

The 2 negative reviewers have main concerns in: limited baseline comparisons and limited testing databases (from both negative reviewers), experimental setup, complicated formulations and no clear takeaways, limited discussion of limitations, etc.
These 2 reviewers have not participated in the discussions.
Though some of the concerns are addressed during the rebuttal (e.g., adding two baselines, clarifications for experimental setup), some are still not addressed. The authors include two more baselines for comparison, but the experimental verification is still not sufficient, and the testing databases are still limited. Overall the quantitate experimental results are still limited in the paper.

**Reviewer Concerns:**

See the above summary.

**Reviewer Scores:**

See the above summary.

---

### Decision · Program_Chairs · 2026-01-26

Reject